# Staphylococcal superantigen-like protein 10 induces necroptosis through TNFR1 activation of RIPK3-dependent signal pathways

Nan Jia[1,2], Guo Li[3], Xing Wang[4], Qing Cao[5], Wanbiao Chen[1], Chengliang Wang[1], Ling Chen[1], Xiaoling Ma[1], Xuan Zhang[1], Yue Tao [2✉], Jianye Zang [1✉], Xi Mo [2✉] & Jinfeng Hu [2,3✉]

*Staphylococcal aureus* (*S. aureus*) infection can lead to a wide range of diseases such as sepsis and pneumonia. Staphylococcal superantigen-like (SSL) proteins, expressed by all known *S. aureus* strains, are shown to be involved in immune evasion during *S. aureus* infection. Here, we show that SSL10, an SSL family protein, exhibits potent cytotoxicity against human cells (HEK293T and HUVEC) by inducing necroptosis upon binding to its receptor TNFR1 on the cell membrane. After binding, two distinct signaling pathways are activated downstream of TNFR1 in a RIPK3-dependent manner, i.e., the RIPK1-RIPK3-MLKL and RIPK3-CaMKII-mitochondrial permeability transition pore (mPTP) pathways. Knockout of *ssl10* in *S. aureus* significantly reduces cytotoxicity of the culture supernatants of *S. aureus*, indicating that SSL10 is involved in extracellular cytotoxicity during infection. We determined the crystal structure of SSL10 at 1.9 Å resolution and identified a positively charged surface of SSL10 responsible for TNFR1 binding and cytotoxic activity. This study thus provides the description of cytotoxicity through induction of necroptosis by the SSL10 protein, and a potential target for clinical treatment of *S. aureus*-associated diseases.

[1] Department of Clinical Laboratory, The First Affiliated Hospital of USTC, Division of Life Sciences and Medicine, University of Science and Technology of China, Hefei 230026 Anhui, China. [2] The Laboratory of Pediatric Infectious Diseases, Pediatric Translational Medicine Institute, Shanghai Children's Medical Center, Shanghai Jiao Tong University School of Medicine, Shanghai 200127, China. [3] Fujian Key Laboratory of Translational Research in Cancer and Neurodegenerative Diseases, Institute for Translational Medicine, School of Basic Medical Sciences, Fujian Medical University, Fuzhou 350122, China. [4] Department of Laboratory Medicine, Shanghai Children's Medical Center, Shanghai Jiao Tong University School of Medicine, Shanghai 200127, China. [5] Department of Infectious Diseases, Shanghai Children's Medical Center, Shanghai Jiao Tong University School of Medicine, Shanghai 200127, China. ✉email: taoyue@scmc.com.cn; zangjy@ustc.edu.cn; xi.mo@shsmu.edu.cn; hujinfeng@fjmu.edu.cn

Staphylococcus aureus (S. aureus) is a prevalent and opportunistic pathogen that causes a wide range of diseases such as sepsis, pneumonia, endocarditis, and osteomyelitis, threatening the health of both humans and animals[1]. Moreover, S. aureus is among the most clinically challenging pathogens worldwide because of its propensity for rapid development and sharing of antibiotic resistance[2]. Although antibiotic treatments can reduce the case fatality rate of most S. aureus related diseases, high mortality rates have still been reported for some severe infectious diseases. For example, the case fatality rate for S. aureus bacteremia can range between 15 and 50%[3]. However, the mechanisms underlying these poor outcomes for some S. aureus-induced diseases have remained largely unknown.

S. aureus can manipulate the host immune response through the expression of a myriad of virulence factors responsible for tissue adherence, immune evasion, cell injury, and organ failure, ultimately promoting its survival and pathogenesis[2,4,5]. The activities of some virulence factors have been shown to induce host cell death, especially through apoptosis and necroptosis, which facilitates immune evasion and tissue damage[6,7]. Necroptosis is a programmed form of necrosis that is regulated in a RIPK3 kinase signaling-dependent manner[8]. Several types of receptors participate in the initiation stage of necroptosis, including TNF superfamily receptors, toll-like receptors (TLR3 and TLR4), and interferon receptors[9]. Among these receptors, TNFR1 is well characterized for its role in triggering caspase-independent cell death via activation of RIPK1 and RIPK3 when stimulated by TNFα[10,11]. MLKL is an important downstream effector of RIPK3 due to its role in the formation of permeable cell membrane channels that lead to cell death[12]. In addition, CaMKII is also phosphorylated by RIPK3, resulting in the opening of mitochondrial permeability transition pores (mPTPs) and subsequent necroptosis in cardiomyocytes, independent of MLKL[13].

Several virulence factors secreted by S. aureus have been shown to induce necroptosis of the host immune cells. For example, S. aureus toxins including Hla, PSM, LukAB, and PVL can induce RIPK1/RIPK3/MLKL-dependent necroptosis in macrophages[7], while PSMα is also reported to trigger neutrophil necroptosis mediated by MLKL[14]. In addition to virulence factors, phagocytosis of S. aureus has been demonstrated to elicit necroptosis of neutrophils by activating RIPK3 in an MLKL-independent manner[15,16], and S. aureus small colony variants can affect host trained immunity by inducing host cell necroptosis via activating glycolysis[17]. In severe sepsis caused by S. aureus infection, vascular permeation, immunosuppression, and organ failure are usually present, which strongly suggests the occurrence of cell death including necroptosis[5,18,19]. However, the detailed mechanism by which S. aureus or its secreted toxic proteins induce necroptosis in non-immune cells is relatively unknown.

Staphylococcal superantigen like (SSL) proteins comprise a family of 14 member proteins with sequences and structures homologous to superantigens but lacking superantigen activities. The 14 ssl genes are encoded at two different loci, with ssl1-ssl11 in the genomic island vSaα, and ssl12-ssl14 in the immune evasion cluster 2[20]. Among all the SSL proteins, SSL10 is a well-studied member that has been found in most human and animal isolates of S. aureus[20–24] and involved in several pathological processes. For example, SSL10 can bind to CXCR4 and then inhibit migration of leukemia cells[25]. SSL10 also blocks the interaction between IgG and complement component C1q that consequently prevent the activation of the classical complement pathway[24,26]. Furthermore, SSL10 interacts with prothrombin and factor Xa to impair blood coagulation[27]. It has also been shown that SSL10 could interfere with host cell inflammation via binding to

ERK2[28]. All these studies suggest that SSL10 possesses multiple functions during S. aureus infection, highlighting the importance of SSL10.

Although SSL proteins have diverse functions in modulating host response to S. aureus infection, it remains unknown whether SSL family proteins can induce cytotoxicity. In the present study, we demonstrate that SSL10 exhibits potent cytotoxicity toward HEK293T and HUVEC cell growth by inducing cellular necroptosis and contributes to the cytotoxicity induced by S. aureus. Mechanistically, SSL10 induces necroptosis through binding to TNFR1 and activating two distinct signal pathways downstream RIPK3. This study provides evidence that SSL10 is a cytotoxic virulence factor that may serve as a therapeutic target in S. aureus infections.

## Results

**SSL10 induces cell necrosis**. To determine the cytotoxicity of SSLs on host cells, we cloned and expressed all of the 14 SSL members in E. coli. SSL3, SSL7, SSL8, SSL10, and SSL11 proteins were successfully purified with high quality for cytotoxicity assays (Supplementary Methods). Human umbilical vein endothelial cells (HUVEC) were treated with these purified recombinant SSL proteins for different time periods. As determined by MTS assay (Supplementary Fig. 1), SSL10, but not SSL3, SSL7, SSL8 or SSL11, significantly reduced the cell activity of HUVEC after 2-day treatment. Furthermore, SSL10 treatment decreased cell viability in a dose- and time-dependent manner in human embryonic kidney cells (HEK293T) and HUVEC (Fig. 1a). Most non-viable cells were PI-positive but Annexin V-negative when detected by flow cytometry, suggesting that cell death induced by SSL10 is most likely to be necrosis (Fig. 1b, c and Supplementary Fig. 2).

Cell necrosis was further confirmed by transmission electron microscopy in which the cells exhibited a typical necrotic phenotype, including cytoplasmic lightening, swollen organelle, and membrane rupture (Fig. 1d). In addition, pretreatment with pan-caspase inhibitor Z-VAD-fmk had little effect on SSL10-induced LDH release, indicating that SSL10 induces necrosis but not apoptosis or pyroptosis (Fig. 1e).

Because SSL10 is a virulence factor secreted by S. aureus, we also detected the effects of SSL10 in the supernatants of S. aureus. We quantified the cytotoxicity of supernatants from wild type (WT) S. aureus strain (NCTC 8325), its ssl7 knockout, ssl10 knockout, or ssl10 complementation strain toward HEK293T and HUVEC. Compared with medium-treated cells, LDH release was significantly increased in cells treated with supernatants from the WT S. aureus. ssl10 knockout but not ssl7 knockout significantly hampered the LDH release induced by S. aureus supernatant, which can be rescued by ssl10 complementation to a level similar to WT S. aureus (Fig. 1f). All these data indicate that SSL10 from S. aureus can induce cell necrosis of both HEK293T and HUVEC.

**SSL10 induces necroptosis via the RIPK3-dependent pathway**. To explore the underlying mechanism by which SSL10 induced necrosis, we pretreated the HEK293T or HUVEC cells with a specific RIPK1 inhibitor (Nec-1s) or RIPK3 inhibitor (GSK'872) prior to SSL10 exposure. As determined by LDH release, Nec-1s could partially attenuate the effects of SSL10, while GSK'872 almost completely inhibited the necrotic effects of SSL10 (Fig. 2a). These data suggest that SSL10 may induce RIPK3-dependent cellular necroptosis.

To further test this hypothesis, we generated knockout cell lines for key genes involved in the necroptosis pathway via CRISPR-Cas9 in both HEK293T and HUVEC cells (Supplementary Fig. 3 and Table 1). Consistent with the effects of inhibitor treatment, knockout of RIPK3 almost completely inhibited SSL10-induced

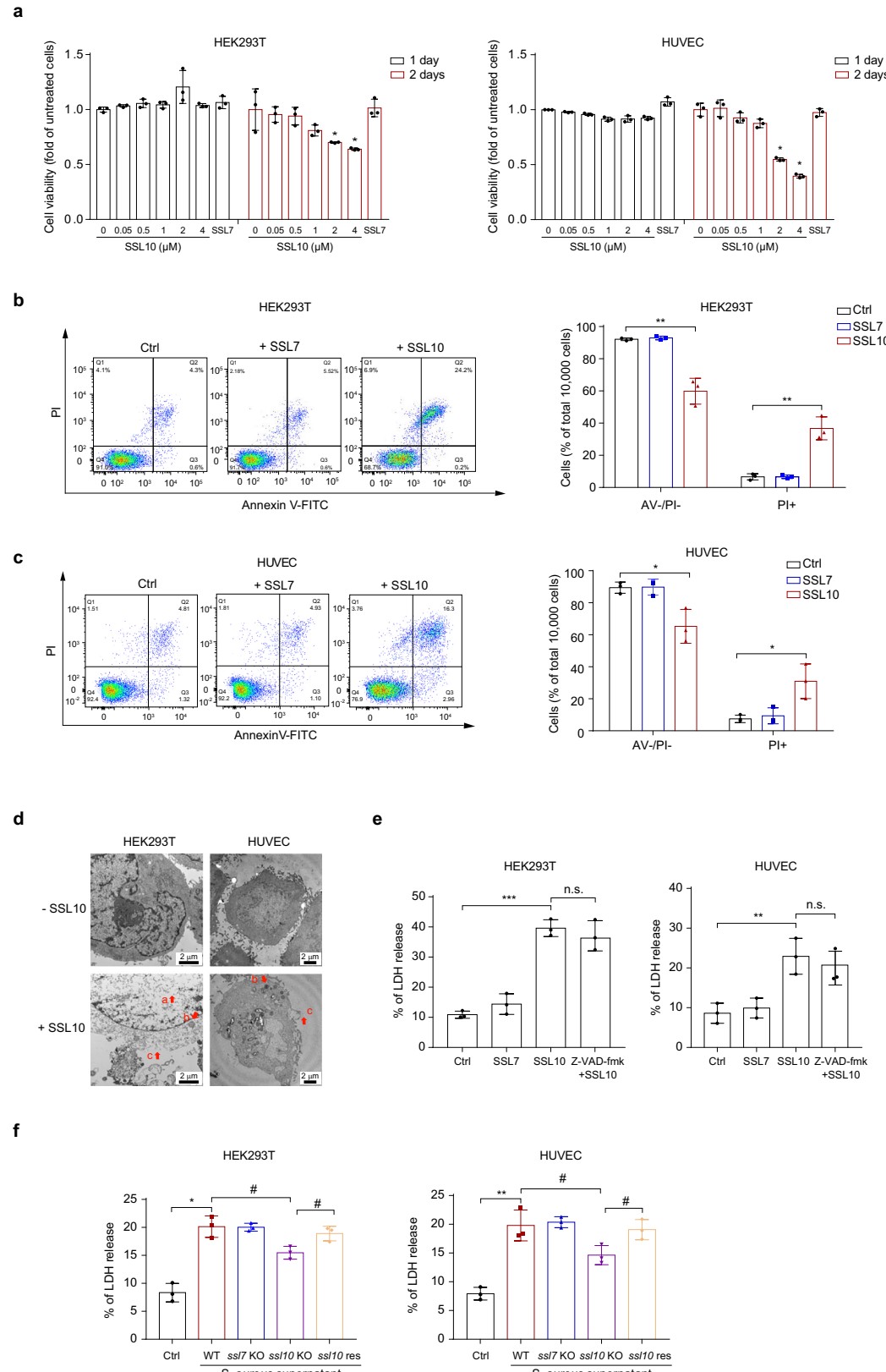

necrosis, while knockout of RIPK1 or MLKL only resulted in partial inhibition (Fig. 2b–d). In addition, transient complementation with RIPK3 in $Ripk3^{-/-}$ HEK293T cells led to robust necrosis, evidenced by the release of LDH (Fig. 2e). Together, these data indicate that SSL10 could induce RIPK3-dependent necroptosis in both HEK293T and HUVEC.

**CaMKII activation and mPTP opening also contribute to SSL10-induced necroptosis.** Previous studies have reported that RIPK1 can form a complex with RIPK3, which further activates MLKL, resulting in necroptosis of several types of cells[11,29]. However, in the present study, we found that inhibition or knockout of RIPK1 or MLKL could not completely inhibit SSL10-

**Fig. 1 SSL10 induces necrosis in HEK293T and HUVEC cells. a** HEK293T and HUVEC cells were treated with different concentrations of SSL10 for different time periods, as indicated. The cell viability was determined by MTS assay. HEK293T (**b**) and HUVEC (**c**) cells were collected after treatment with 2 μM SSL10 for 48 h at 37 °C and 5% $CO_2$, and then detected by flow cytometry using Annexin V/PI staining. The dot plot (left) is representative of three independent experiments, and the quantification results are shown as a bar graph (right). **d** Transmission electron microscopy images of HEK293T and HUVEC cells treated with or without 2 μM SSL10 for 48 h at 37 °C and 5% $CO_2$. Red letters with arrows indicate characteristic features of necrotic morphology: **a** cytoplasmic lightening; **b** swollen organelle; **c** membrane rupture. **e** Cells were pretreated with 10 μM Z-VAD-fmk for 30 min at 37 °C and 5% $CO_2$, and then stimulated with 2 μM SSL10. The release of LDH was detected. **f** The supernatant of wild type *S. aureus* NCTC 8325, *ssl10* knockout, *ssl7* knockout or *ssl10* complementation bacteria was used to treat HEK293T and HUVEC cells for 48 h at 1:10 dilution at 37 °C and 5% $CO_2$. LDH released from the cells was evaluated. WT wild type. Cells treated with 2 μM SSL7 were used as a negative protein control throughout the experiments. All data represent the means ± SD calculated from three independent experiments. *$p < 0.05$; **$p < 0.01$; ***$p < 0.001$ compared to the Ctrl cells (buffer-treated cells or TSB medium-treated cells). #$p < 0.05$ compared to the WT or *ssl10* knockout *S. aureus* supernatant treated group as indicated. n.s. not significant, by one-way ANOVA (**b**–**f**) or two-way ANOVA (**a**).

induced necroptosis, suggesting that SSL10-induced necroptosis may also depend on other RIPK3-mediated pathways independent of RIPK1 and MLKL.

In addition to MLKL, RIPK3 has been reported to phosphorylate CaMKII to induce the opening of mPTP channels, for example, leading to necroptosis in cardiomyocytes[13]. To explore whether CaMKII was involved in SSL10-induced necroptosis, HEK293T and HUVEC cells were treated with KN-93, a selective inhibitor of CaMKII prior to SSL10 treatment. As assessed by the release of LDH and ATP, inhibition of CaMKII profoundly abrogated SSL10-induced necroptosis (Fig. 3a, b). Phospho-CaMKII levels were also significantly increased after SSL10 treatment (Fig. 3c), which suggested the involvement of CaMKII in SSL10-induced necroptosis.

To further identify the downstream effector of CaMKII, we pretreated HEK293T and HUVEC cells with CsA, an inhibitor of mPTP opening, and found that CsA efficiently blocked SSL10-induced LDH release (Fig. 3d). In addition, SSL10 treatment led to mitochondrial depolarization, which was significantly hampered in the absence of RIPK3, evidenced by the recovery in mitochondrial membrane potential ($\Delta\Psi_m$) (Fig. 3e). These data indicate that CaMKII-mPTP is likely to be a primary candidate downstream of RIPK3 in SSL10-induced necroptosis.

**SSL10 induces necroptosis by direct interaction with the TNFR1 extracellular domain (TNFR1$^{ECD}$).** Necroptosis is initiated through ligand binding to several receptors including TNFR1, toll-like receptors (TLR3 and TLR4), and interferon receptors[30–32]. To explore whether SSL10 induces necroptosis by interacting with membrane receptors, SSL10 localization was examined by real-time live-cell analysis and scanning confocal microscopy (Supplementary Methods). Notably, GFP-tagged SSL10 was found to be enriched on the HUVEC cell membrane within the first 30 min of treatment, suggesting that SSL10 may bind to a cell surface receptor (Supplementary Fig. 4). Similarly, His-tagged SSL10 proteins, but not SSL7 proteins, were found on the HEK293T cell surface as detected by flow cytometry (Fig. 4a).

To further determine which membrane receptor was involved in SSL10-induced necroptosis, inhibitors against TLR4 (TAK-242) and interferon receptor IFNAR1 (IFN alpha-IFNAR-IN-1) were used to pre-treat HEK293T cells before SSL10 exposure. However, SSL10-induced necroptosis was not affected by these inhibitors, indicating that SSL10-induced necroptosis is independent of TLR4 or IFNAR1 (Supplementary Fig. 5). In addition to these two receptors, TNFR1 is a well characterized receptor that can trigger necroptosis through activating RIPK3[9]. To test whether TNFR1 participated in SSL10-induced necroptosis, we knocked out TNFR1 in HEK293T and HUVEC cells via CRISPR-Cas9 and found that depletion of TNFR1 significantly decreased the binding of SSL10 to the cells (Fig. 4a), and blocked SSL10-induced necroptosis, as indicated by the significantly decreased

release of LDH (Fig. 4b). Significantly, increased viable cell counts and $\Delta\Psi_m$ were found in TNFR1-depleted cells after SSL10 treatment (Fig. 4c, d). In addition, anti-TNFR1 antibody and purified TNFR1$^{ECD}$, the extracellular domain of human TNFR1 containing amino acids 22-211 (GenBank number CAA39021.1), inhibited SSL10-induced necroptosis in a dose-dependent manner in HUVEC (Supplementary Fig. 6a, b). These data suggest that TNFR1 is involved in SSL10-induced necroptosis, likely serving as the cellular receptor for SSL10.

To further test whether TNFR1 was the receptor for SSL10, we employed in vitro MBP pull-down assays using purified His-tagged SSL10, or His-tagged SSL7 with MBP-tagged TNFR1$^{ECD}$, and found that SSL10, but not SSL7, interacted with TNFR1$^{ECD}$ (Fig. 4e). Consistently, as determined by isothermal titration calorimetry (ITC) assays, the $K_D$ value between SSL10 and TNFR1$^{ECD}$ was $3.87 \pm 0.11$ μM, while there was no binding between SSL7 and TNFR1$^{ECD}$ (Fig. 4f), suggesting a specific binding of SSL10 to TNFR1. Taken together, these data demonstrate that SSL10 activates cell necroptosis via direct interaction with TNFR1$^{ECD}$.

**Overall structure of SSL10.** To further understand the molecular mechanisms driving the SSL10 activation of necroptosis via binding to TNFR1, we next determined the crystal structure of SSL10 by molecular replacement at 1.9 Å resolution. X-ray diffraction data and structure refinement statistics are shown in Table 2. SSL10 existed as a monomer in both solution and crystal structure (Supplementary Fig. 7). Two SSL10 molecules were observed in one asymmetric unit adopting approximately identical structures, with the RMSD value of 0.254 Å when the two molecules were aligned (Supplementary Fig. 8a). In light of these results, we selected molecule B for further investigation. SSL10 exhibited a typical superantigen-like structure, consisting of two distinct domains separated by a flexible linker region. The N-terminal OB-fold domain (residues 43–123) contains one α-helix, eight β-strands, and one $3_{10}$ helix, while the C-terminal β-grasp domain (residues 133–227) consists of one α-helix, seven β-strands, and two $3_{10}$ helices (Fig. 5a). Structural comparison of SSL10 with other SSL proteins (i.e., SSL3, SSL4, SSL5, SSL7, SSL8, and SSL11) revealed that the SSL proteins share similar folds (Supplementary Fig. 8b) but are quite different in electrostatic surface potential (Fig. 5b and Supplementary Fig. 8c), which might be responsible for the diverse binding partners and functions of the SSL proteins.

**Both the N- and C-terminal domains of SSL10 contribute to its cytotoxicity.** To investigate which domain or domains of SSL10 may be critical for its cytotoxicity, we generated variants of SSL10 with the N- and C-terminal domains swapped between SSL7. We designated the two newly generated chimeric proteins as

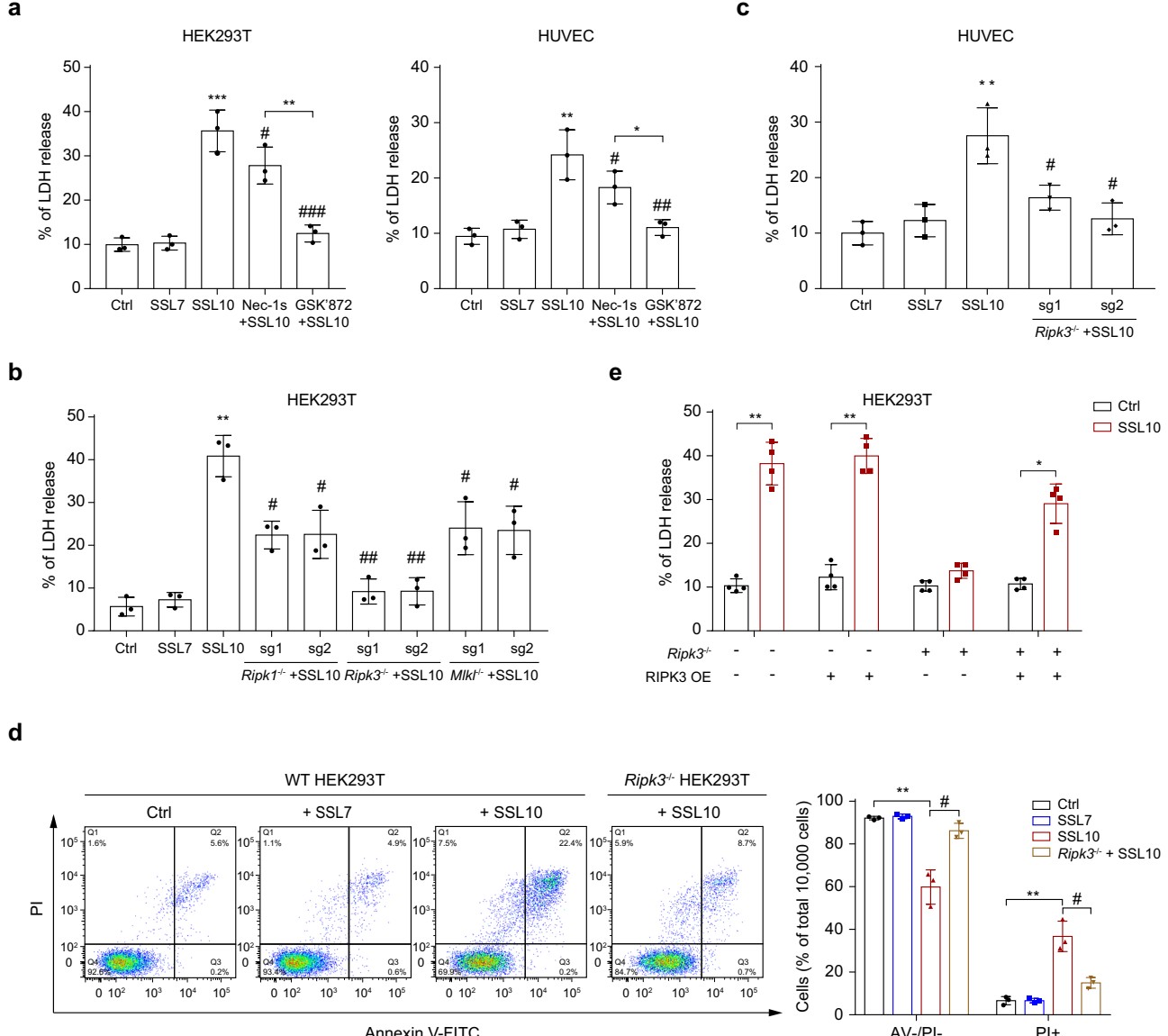

**Fig. 2 SSL10 triggers RIPK3-dependent necroptosis of HEK293T and HUVEC cells. a** HEK293T and HUVEC cells were pretreated with 50 μM Nec-1s or 50 μM GSK'872 for 30 min and then stimulated with 2 μM SSL10 for 48 h at 37 °C and 5% $CO_2$. The release of LDH was detected. **b** *Ripk1*, *Ripk3*, or *Mlkl* knockout (KO) HEK293T cells were treated with 2 μM SSL10 for 48 h at 37 °C and 5% $CO_2$, and the LDH release was detected. **c** Wild type (WT) or *Ripk3* KO HUVEC were treated with 2 μM SSL10 for 48 h at 37 °C and 5% $CO_2$, and the LDH release was detected. **d** WT or *Ripk3* KO HEK293T cells were treated with SSL10 for 48 h and the cell viability was measured by flow cytometry after Annexin V/PI staining. The dot plot (left) is representative of three independent experiments, and the results of quantification (right) are shown as a bar graph. **e** WT or *Ripk3* KO HEK293T cells with or without transient complementation of *Ripk3* were treated with 2 μM SSL10 for 48 h at 37 °C and 5% $CO_2$, and the LDH release was detected. Cells treated with 2 μM SSL7 were used as a negative protein control throughout the experiments. All data are presented as the means ± SD calculated from three independent experiments. *$p < 0.05$; **$p < 0.01$; ***$p < 0.001$ compared to the Ctrl cells (buffer-treated cells). #$p < 0.05$; ##$p < 0.01$; ###$p < 0.001$ compared to the SSL10 treated group, by one-way (**a** and **d**) or two-way ANOVA (**b**, **c**, and **e**).

| Table 1 Sequences of sgRNA for CRISPR-Cas9 genome editing. | | | |
|---|---|---|---|
| **Genes** | **sgRNA** | **Oligo 1 (5′-3′)** | **Oligo 2 (5′-3′)** |
| *Ripk1* | *sgRNA-1* | 5′-GACGTGAAGAGTTTAAAGGT-3′ | 5′-ACCTTTAAACTCTTCACGTC-3′ |
| | *sgRNA-2* | 5′-AGTACTCCGCTTTCTGTAAA-3′ | 5′-TTTACAGAAAGCGGAGTACT-3′ |
| *Ripk3* | *sgRNA-1* | 5′-TTCAGCAGGCGGCAAAGGAG-3′ | 5′-CTCCTTTGCCGCCTGCTGAA-3′ |
| | *sgRNA-2* | 5′-GGACCCAGAGCTGCACGTCA-3′ | 5′-TGACGTGCAGCTCTGGGTCC-3′ |
| *Mlkl* | *sgRNA-1* | 5′-GCTTGATCAGGCCGAGGACG-3′ | 5′-CGTCCTCGGCCTGATCAAGC-3′ |
| | *sgRNA-2* | 5′-GCATCTCCAGAGGCTTGATC-3′ | 5′-GATCAAGCCTCTGGAGATGC-3′ |
| *Tnfrsf1a* | *sgRNA-1* | 5′-GGTGGGAATATACCCCTCAG-3′ | 5′-CTGAGGGGTATATTCCCACC-3′ |
| | *sgRNA-2* | 5′-GGTGGCACCACCCTATCAGG-3′ | 5′-CCTGATAGGGTGGTGCCACC-3′ |

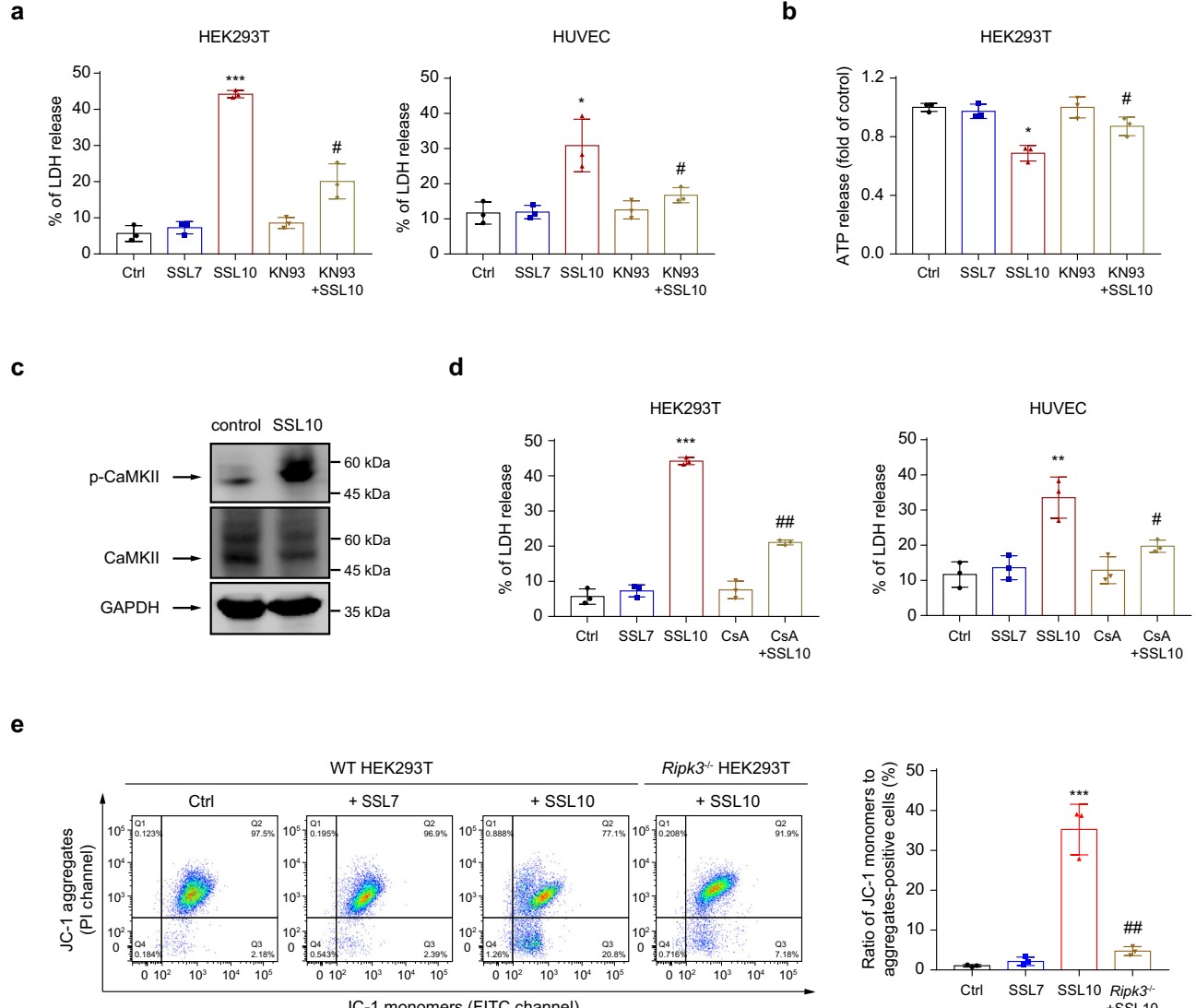

**Fig. 3 CaMKII activation and mPTP opening are essential for SSL10-induced necroptosis in HEK293T and HUVEC cells.** HEK293T or HUVEC cells were pretreated with 5 µM KN-93 for 30 min before SSL10 treatment, and release of LDH (**a**) and ATP (**b**) was determined. **c** CaMKII and phospho-Thr287 CaMKII were detected by immunoblotting in whole cell lysates of HEK293T cells treated with or without SSL10. **d** HEK293T or HUVEC cells were pretreated with 1 µM CsA for 30 min before SSL10 treatment, and the LDH release was detected. **e** Depolarization of the mitochondrial membrane of wild type (WT) HEK293T cells or *Ripk3* knockout cells treated with SSL10 was measured by flow cytometry after JC-1 staining. The dot plot (left) is representative of three independent experiments, and the results of quantification are shown as a bar graph (right). Cells treated with 2 µM SSL7 were used as a negative protein control throughout the experiments. All data are presented as the means ± SD calculated from three independent experiments. *$p < 0.05$; **$p < 0.01$; ***$p < 0.001$ compared to the Ctrl cells (buffer-treated cells). #$p < 0.05$; ##$p < 0.01$ compared to the SSL10 treated group, by one-way (**a**–**d**) or two-way ANOVA (**e**).

SSL7 × 10 and SSL10 × 7, with SSL7 × 10 containing the SSL7 N-terminus and the SSL10 C-terminus, and vice versa (Fig. 5c). We found that both of the two chimeric proteins induced a marked release of LDH, but were significantly less potent than SSL10 (Fig. 5d). In agreement with these results, both of the chimeric proteins could still bind to TNFR1ECD with weaker binding activity compared to SSL10 (Fig. 5e). These data indicate that both the N- and C- terminal domains of SSL10 interact with TNFR1ECD and participate in SSL10-induced necroptosis.

**Predicted binding interface between SSL10 and TNFR1.** To analyze the regions in TNFR1ECD mediating its interaction with SSL10, we constructed 4 MBP-tagged TNFR1 deletion mutants (i.e., ΔCRD1, ΔCRD2, ΔCRD3, and ΔCRD4) according to its four

extracellular CRDs[33]. We found that ΔCRD2 mutant of TNFR1ECD, but not the other three mutants, showed significantly decrease in its binding ability to SSL10 (Fig. 6a). Such decrease in the interaction between ΔCRD2 mutant of TNFR1ECD and SSL10 was further confirmed by the decreased binding affinity (with $K_D$ value of 30.3 ± 3.55 µM) as determined by ITC (Fig. 6b). Consistently, compared to WT TNFR1ECD, the inhibitory effects to SSL10 induced-cytotoxicity of ΔCRD2 mutant was also significantly decreased (Fig. 6c). These results confirm that the second CRD region of TNFR1ECD is important for its interaction with SSL10.

To further understand the molecular mechanism underlying the interaction between SSL10 and TNFR1, the structure of TNFR1ECD (PDB code: 1EXT) was docked onto the structure of SSL10 using the HawkDock webserver (http://cadd.zju.edu.cn/hawkdock/)[34].

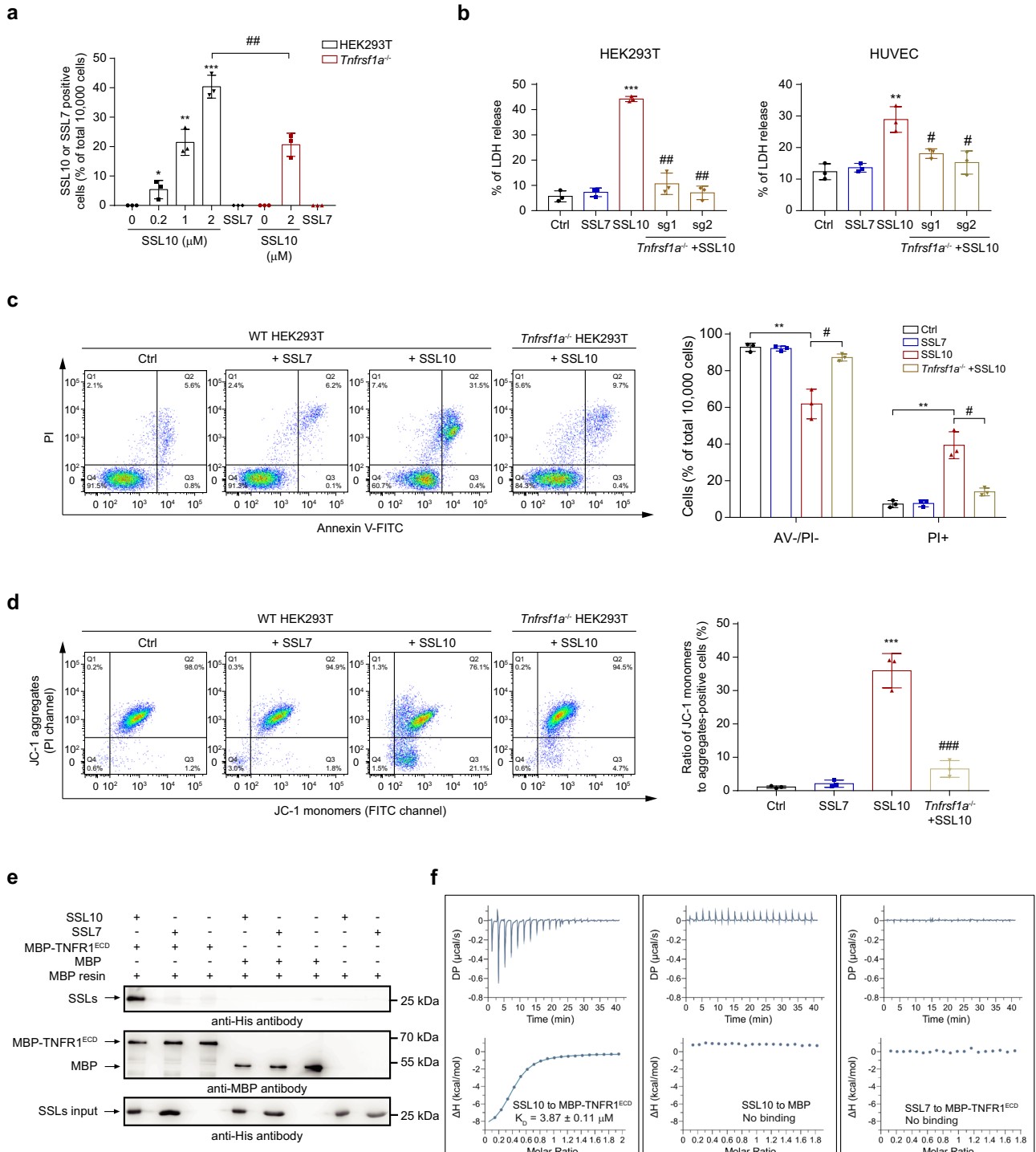

**Fig. 4 SSL10 induces necroptosis by direct interacting with the extracellular domain of TNFR1 (TNFR1^ECD). a** Wild type (WT) or *Tnfrsf1a* knockout (KO) HEK293T cells were treated with SSL10 at different concentrations for 30 min, and the SSL proteins bound to the cell surface were detected by flow cytometry after FITC-conjugated anti-His-tag antibody staining. **b** WT or *Tnfrsf1a* KO cells were treated with 2 μM SSL10 for 48 h at 37 °C and 5% $CO_2$, and the LDH release was detected. **c** WT or *Tnfrsf1a* KO HEK293T cells were challenged with 2 μM SSL10 for 48 h and the cell viability was measured by flow cytometry after Annexin V/PI staining. The dot plot (left) is representative of three independent experiments, and the results of quantification (right) are shown as a bar graph. **d** Depolarization of the mitochondrial membrane of WT or *Tnfrsf1a* KO HEK293T cells treated with SSL10 was measured by flow cytometry after JC-1 staining. The dot plot (left) is representative of three independent experiments, and the results of quantification (right) are shown as a bar graph. **e** Immunoblotting of SSL10 after pull-down with MBP-TNFR1^ECD. **f** ITC assays for SSL10 binding to MBP-TNFR1^ECD or MBP control protein. Cells treated with 2 μM SSL7 were used as a negative protein control throughout the experiments. All data are presented as the means ± SD from three independent experiments. *$p < 0.05$; **$p < 0.01$; ***$p < 0.001$ compared to the Ctrl cells (buffer-treated cells). #$p < 0.05$; ##$p < 0.01$; ###$p < 0.001$ compared to the SSL10 treated group, by two-way ANOVA.

**Table 2 Crystallographic data collection and structure refinement.**

| | SSL10 |
|---|---|
| PDB code | 6LWT[a] |
| Data collection | |
| Space group | $P2_1 2_1 2_1$ |
| Cell dimensions | |
| a, b, c (Å) | 41.24, 71.21, 140.66 |
| α, β, γ (°) | 90, 90, 90 |
| Resolution (Å) | 50.0–1.90 (1.93–1.90)[b] |
| Wavelength (Å) | 0.9778 |
| $R_{sym}$ or $R_{merge}$ (%) | 9.0 (29.4) |
| Overall $I/\sigma(I)$ | 16.2 (7.0) |
| Completeness (%) | 100.0 (100.0) |
| Redundancy | 6.4 (6.6) |
| Refinement | |
| Resolution (Å) | 39.58–1.90 |
| No. reflections | 31,821 |
| $R_{work}/R_{free}$ (%) | 20.94/24.62 |
| No. atoms | |
| Protein/Water | 3127/57 |
| Average B-factors (Å$^2$) | |
| Protein/Water | 30.858/27.255 |
| R.m.s deviations | |
| Bond lengths (Å) | 0.0088 |
| Bond angles (°) | 1.3524 |
| Ramachandran plot[c] | |
| Most favored regions (%) | 96.23 |
| Allowed regions (%) | 3.77 |
| Generously allowed regions (%) | 0 |

[a]One single crystal of SSL10 was used for the structure determination.
[b]The values in parentheses refer to the highest resolution shell.
[c]Statistics for the Ramachandran plot from an analysis using MolProbity.

The binding interfaces of the top 10 predicted models were further analyzed by Molecular Mechanics/Generalized Born Surface Area (MM/GBSA) method[35]. The binding free energies of these top 10 models ranged from −37.63 kcal/mol to −7.51 kcal/mol (Supplementary Table 1). As shown previously, both the N- and C-terminal domains of SSL10 contribute to TNFR1$^{ECD}$ interaction and necroptosis induction (Fig. 5), whereas the recognition of SSL10 is largely mediated by CRD2 of TNFR1$^{ECD}$ (Fig. 6a–c). Therefore, the models with the interface formed by both the N- and C-terminal domain of SSL10 and the CRD2 region of TNFR1$^{ECD}$, which were model 1, 2, 4, and 8, were included for further analysis (Supplementary Fig. 9a). In each model, TNFR1$^{ECD}$ binds to a positively charged region on SSL10, suggesting the recognition of TNFR1 by SSL10 is dominantly mediated by electrostatic interactions (Fig. 6d, e and Supplementary Fig. 9b).

To assess the reliability of these models, four SSL10 mutants, named M1, M2, M4, and M8 were generated by replacing the important residues buried in the interface by amino acid residues either with the opposite charged or the corresponding counterparts present in SSL3, SSL7, SSL8, or SSL11 (Supplementary Table 2, Figs. 6d, e, S9b and S10a). Indicated by size exclusion chromatography, we obtained aggregated M4 and M8 proteins which were improperly folded (Supplementary Fig. 10b). Therefore, we used the monomeric M1 and M2 for MBP pull-down assays. As shown in Fig. 6f, the binding of M1 to TNFR1$^{ECD}$ was abolished, while M2 showed comparable binding ability to TNFR1$^{ECD}$ as WT SSL10. Disruption of the interaction between M1 and TNFR1$^{ECD}$ was further confirmed by the ITC assay (Fig. 6g). Consistently, the release of LDH from HEK293T cells treated by M1 was significantly decreased compared to WT SSL10

(Fig. 6h). These results suggested a binding surface on SSL10 responsible for TNFR1 interaction and necroptosis induction.

## Discussion

Previous studies on SSL10 suggested that SSL10 contributed to *S. aureus* pathogenicity by targeting different cellular responses, including the classical complement activation pathway, the migration of T cells, the interaction between complement C1q and IgG, and the Fc-receptor-mediated phagocytosis of neutrophils[24–27]. In addition, recent publication showed that SSL10 binds to ERK2 to interfere with host cell inflammation[28]. Here, our data demonstrated the cytotoxicity of SSL10 by triggering necroptosis via activation of two distinct signaling pathways upon binding to TNFR1 in HEK293T and HUVEC cells.

SSL10, like other SSL protein family members, adopts conserved structure with an N-terminal OB-fold domain and a C-terminal β-grasp domain[36]. SSL10 has been shown to interact with various partners on distinct binding sites to perform different functions. For example, SSL10 binds to IgG1 predominantly via the N-terminal OB-fold domain to inhibit the FcR-mediated phagocytosis[24], while both OB-fold and β-grasp domains contribute to the interaction of SSL10 with the λ-carboxyglutamic acid (Gla) domain of prothrombin[27,37]. In this study, we identified a TNFR1 binding site on SSL10 surface involving both OB-fold and β-grasp domains based on the docking model of SSL10-TNFR1$^{ECD}$ complex, as well as the mutagenesis and cytotoxic analysis (Fig. 6d–h). Interestingly, the binding site for TNFR1$^{ECD}$ of SSL10 overlap with the binding site (amino acid residues 54-81 and 195-227) to prothrombin[37]. LDH assays reveal that the Gla domain of prothrombin inhibits the cytotoxicity of SSL10 in a dose-dependent manner (Supplementary Fig. 6c). Therefore, under certain circumstances, prothrombin might compete with TNFR1 for interaction with SSL10.

Damage of endothelial cells can facilitate the spread of *S. aureus* into the bloodstream, leading to the development of septic shock and organ failure[38,39]. Virulence factors from *S. aureus* play important roles in the development and progression of sepsis through multiple mechanisms including disrupting various types of host cells[4,6,40–43]. Reijer et al. characterized the serial levels of IgG and IgA antibodies against 56 staphylococcal antigens in multiple serum samples of 21 patients with a *S. aureus* bacteremia. Their data showed that an increase in IgG levels against SSL10 was observed at some time point after the onset of bacteremia in 95 to 100% of all patients[44]. Hemostatic abnormalities frequently occur during sepsis and are most often attributed to disseminated intravascular coagulation. Draaijers et al. reported the case of a patient with severe coagulopathy acquired during fulminant *S. aureus* sepsis and speculated that inhibition of coagulation factor X by *S. aureus* SSL10 is the most likely cause of the acquired coagulopathy in their patient[45].

In addition to the previously identified function, our study revealed a new function of SSL10 to trigger necroptosis of endothelial and epithelial cells in a RIPK3-dependent manner. Although RIPK3 is expressed at an extremely low level in endothelial and epithelial cells, including HUVEC, and HEK293T, its importance in vivo cannot be excluded[46,47]. Genetic evidence showed that RIPK3 deficiency leads to reduced endothelial cell permeability or necroptosis, thereby suppressing tumor metastasis[48,49]. Furthermore, RIPK3 can be induced or upregulated under certain conditions, which confers cells sensitive to RIPK3-dependent necroptosis[48]. Notably, we found an SSL10-induced increase in the protein level of RIPK3 in HEK293T and HUVEC cells (Supplementary Fig. 11), indicating the important role of RIPK3 on SSL10-induced necroptosis. In

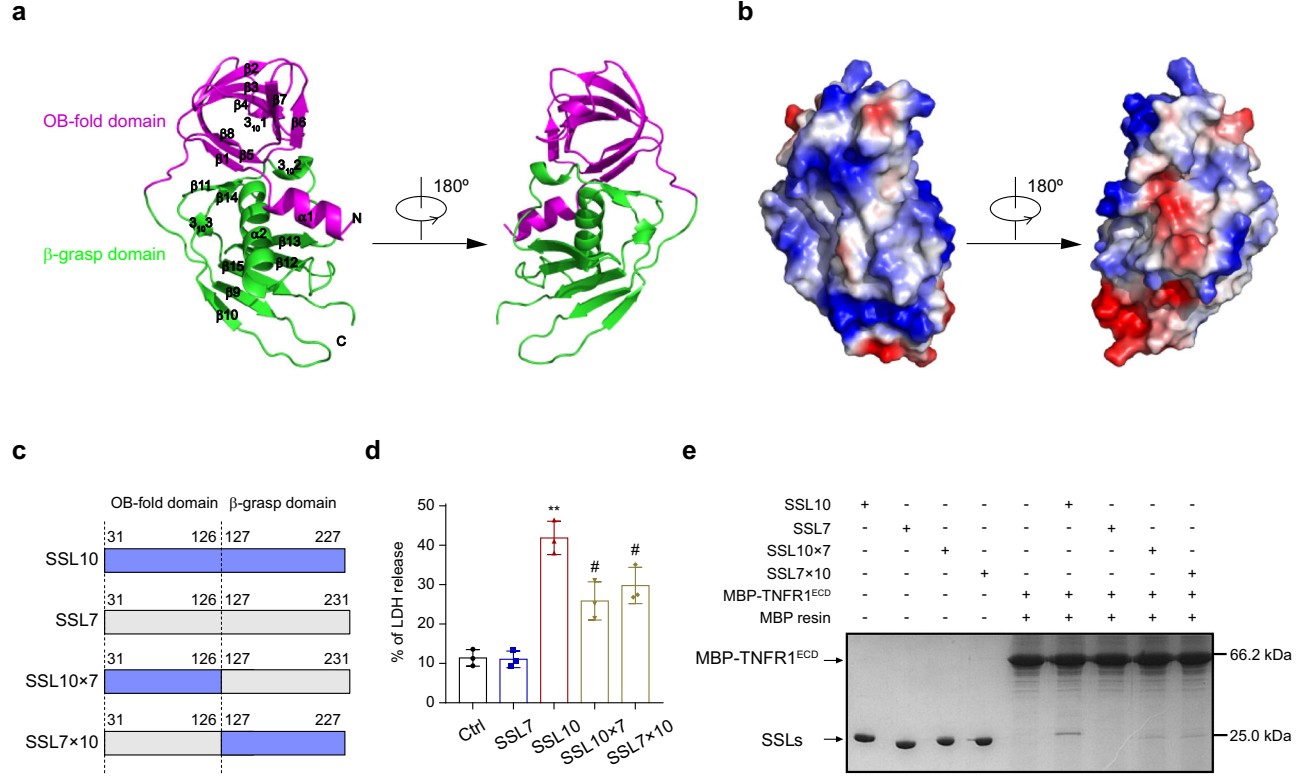

**Fig. 5 Both the N- and C-terminal domains contribute to SSL10-induced necroptosis. a** The overall structure of SSL10. The OB-fold domain and β-grasp domain of SSL10 are colored magenta and green, respectively. **b** The electrostatic surface potentials of SSL10. Positively and negatively charged surface regions are colored blue and red, respectively. **c** Illustration of chimeric SSL variants used in this study. Blue and gray boxes represent sequences derived from SSL10 and SSL7, respectively. The numbers indicate the residue positions of the N- and C-terminus of the swapped domains. **d** LDH released from HEK293T cells treated with 2 μM SSL10, SSL7, SSL10 × 7, and SSL7 × 10. All data are presented as the means ± SD from three independent experiments. *$p < 0.05$; **$p < 0.01$ compared to the Ctrl cells (buffer-treated cells). #$p < 0.05$ compared to the SSL10 treated group, by one-way ANOVA. **e** MBP pull-down of SSL10, SSL7, SSL10 × 7 and SSL7 × 10 by MBP-TNFR1ECD.

addition, we also observed that mRNA levels of *ssl10* in the strains isolated from patients with hypoproteinemia, septic shock or multiple organ dysfunction syndrome (MODS) were significantly higher than that in strains from patients without these diagnoses, whereas the mRNA levels of *ssl7* in these strains were similar (Supplementary Methods, Supplementary Fig. 12). Since hypoproteinemia, septic shock or MODS are the hallmarks of endothelial damage, it is quite possible that SSL10 may be involved in the progression of *S. aureus*-induced bacteremia via different mechanisms, including triggering necroptosis of endothelial cells.

In summary, our study demonstrates that SSL10 triggers a signal cascade leading to necroptosis via its direct interaction with TNFR1. Moreover, this signal cascade is activated in a RIPK3-dependent manner and transduced through two independent signaling pathways (Fig. 7). We also identified a binding interface involving a positively charged surface of SSL10 and the second extracellular cysteine-rich domain of TNFR1. This study, in combination with other previous reports, provides strong evidence that SSL10 contributes to *S. aureus* infection via multiple mechanisms. This study also suggests that SSL10 may serve as a potential reliable biomarker and therapeutic target in *S. aureus*-associated infection and diseases.

## Methods

**Reagents and cell culture**. HUVEC line and human embryonic kidney cell line (HEK293T) were obtained from Cell Bank of Chinese Academy of Sciences (Shanghai, China). The cells were cultured in Dulbecco's Modified Eagle's Medium supplemented with 10% fetal bovine serum and incubated at 37 °C in humidified incubator with 5% $CO_2$. *S. aureus* RN4220 and *S. aureus* 8325 strains were kindly

provided by Dr. Min Li (Renji Hospital, Shanghai Jiao Tong University School of Medicine).

Primary antibodies against RIPK1 (1:1000), RIPK3 (1:1000), MLKL (1:1000), TNFR1 (1:1000), CaMKII (1:1000), phospho-CaMKII (1:1000), and GAPDH (1:1000) were purchased from Cell Signaling Technology (Danvers, MA, USA). Primary antibodies against actin and tubulin were purchased from TransGen Biotechnology (Beijing, China). Primary antibodies against His-tag (1:2000) and MBP-tag (1:1000) were purchased from Affinity Biosciences (China) and Santa Cruz Biotechnology (Dallas, USA), respectively. The secondary antibodies including anti-rabbit IgG H&L (HRP-conjugated) and anti-mouse IgG H&L (HRP-conjugated) were obtained from Beyotime Biotechnology (Shanghai, China).

CellTiter 96® AQueous One Solution Cell Proliferation Assay and CytoTox 96® Non-Radioactive Cytotoxicity Assay (Promega, Madison, WI) were used for MTS [3-(4,5-dimethylthiazol-2-yl)-5-(3-carboxymethoxyphenyl)-2-(4-sulfophenyl)-2H-tetrazolium] and lactate dehydrogenase (LDH) assays, respectively. To measure the ATP concentration in the cells, CellTiter-Glo® Luminescent Viability Assay (Promega, Madison, WI) was used, while FITC Annexin V Apoptosis Detection KIT I (BD, New Jersey, US) was used to detect the cell death by flow cytometry.

Inhibitors used in the present study are as follows: necrostatin-1 stable (Nec-1s, Biovision), GSK'872 (Calbiochem), Z-VAD-fmk (Selleckchem), KN-93 (APExBIO), and cyclosporine A (CsA, APExBIO).

**Cytotoxicity assays**. HUVEC or HEK293T cells were seeded in 96-well plate (100 μl per well) at a density of $1 \times 10^3$ or $1 \times 10^4$ cells/well, 1 day prior to treatment with or without SSL10 resuspended in Opti-MEM reduced serum medium (Thermo Fisher) for 48 h. To detect the effects of inhibitors, the cells were pretreated with various inhibitors 0.5 h before 2 μM SSL10 treatment.

For MTS assay, 20 μl of CellTiter 96® AQueous One Solution reagent was added to each well and incubated for two more hours, and the optical density was measured at 490 nm with a BioTek Synergy/2 microplate reader (BioTek, Winooski, VT).

LDH release was measured according to the manufacturer's manual. Briefly, 50 μl culture media from various treated cells were transferred to a new 96-well flat clear bottom plate, and 50 μl of the CytoTox 96® reagent was added to each sample aliquot and incubated in dark for 30 min at room temperature. To determine the maximum

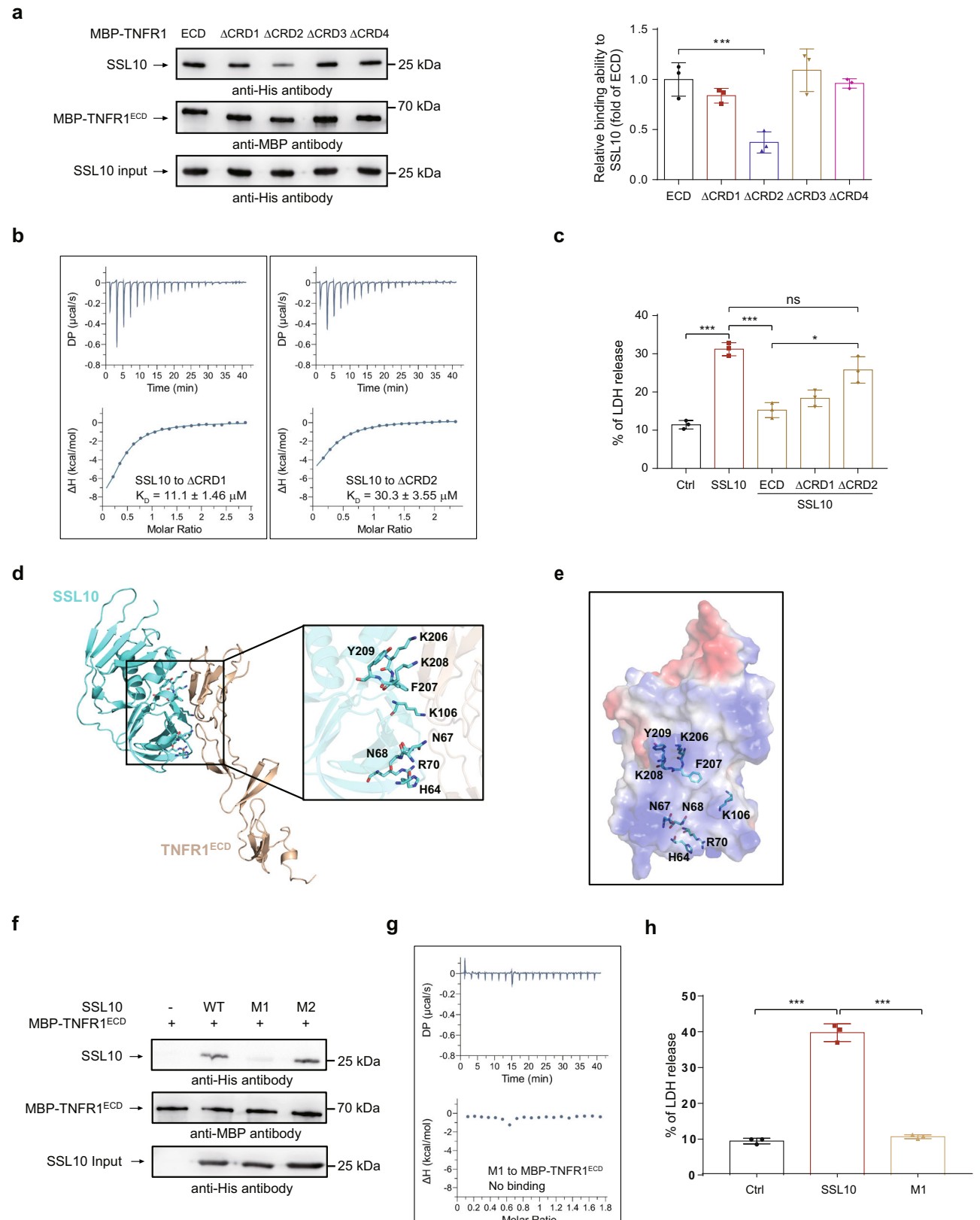

LDH release, 10 µl of 10 × lysis solution was added to 100 µl cell control for 45 min before adding CytoTox 96® reagent. Finally, 50 µl of stop solution was added to each well and the absorbance at 490 nm was recorded with BioTek Synergy/2. LDH leakage was calculated as a percentage of the maximum LDH release of control (buffer-treated cells) group after subtracting the background absorbance.

To determine the ATP concentration in the cells, CellTiter-Glo® reagent was added to each well and the plate was incubated for 10 min to stabilize luminescent signal before luminescence being recorded with BioTek Synergy/2. Luminescent signals from blank wells and buffer-treated cells were used as background and maximal luminescence.

Cell death detected by flow cytometry was performed as previously described[50]. Briefly, 48 h after treatment with SSL10, the cells were collected and washed twice with ice-cold PBS. The cells were then incubated with 5 µl FITC annexin V and 5 µl PI in 500 µl prepared assay buffer in dark for 10 min at room temperature, and

**Fig. 6 Identification of a potential binding interface between SSL10 and TNFR1. a** Immunoblotting of SSL10 after pull-down with MBP-TNFR1$^{ECD}$ or its four deletion mutants (i.e., ΔCRD1-4). The relative band intensities of SSL10 proteins pulled down by WT or mutant MBP-TNFR1$^{ECD}$ are quantitated by densitometry after normalization to their input, and then expressed as the fold of WT SSL10 pulled down by WT MBP-TNFR1$^{ECD}$. **b** ITC assays of SSL10 binding to the TNFR1$^{ECD}$ mutant ΔCRD1 or ΔCRD2. **c** LDH released from HUVEC cells treated with 2 μM SSL10 alone or combined with 10 μM MBP-TNFR1$^{ECD}$, ΔCRD1 or ΔCRD2, as indicated. **d** Model 1 of SSL10/TNFR1$^{ECD}$ complex was generated by HawkDock program with the crystal structures of SSL10 and TNFR1$^{ECD}$ (PDB code: 1EXT). The enlarged view shows the residues of SSL10 predicted to interact with TNFR1$^{ECD}$. SSL10 and TNFR1$^{ECD}$ are colored cyan and light brown, respectively. **e** The electrostatic surface view of predicted binding site of SSL10 for TNFR1$^{ECD}$ is shown. The positive and negative charge are colored blue and red, respectively. **f** Immunoblotting of SSL10 and its mutants (M1 and M2) after pull-down with MBP-TNFR1$^{ECD}$. **g** ITC assay of mutant M1 binding to MBP-TNFR1$^{ECD}$. **h** LDH released from HEK293T cells treated with 2 μM SSL10 or mutant M1. All data represent the means ± SD calculated from three independent experiments. *$p < 0.05$; **$p < 0.01$, ***$p < 0.001$ compared to the Ctrl cells (buffer-treated cells) or as indicated. n.s. not significant, by one-way ANOVA.

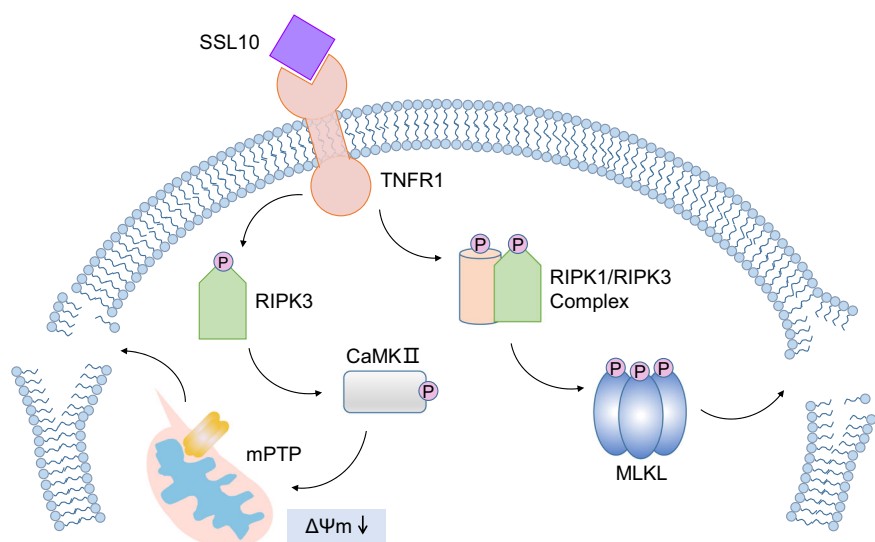

**Fig. 7 Schematic diagram of the SSL10 signal cascade and subsequent cellular effects.** SSL10 secreted by *S. aureus* directly interacts with TNFR1 on the host cell surface and induces cell necroptosis via two distinct pathways, including the RIPK1-RIPK3-MLKL and RIPK3-CaMKII-mPTP pathways.

applied for flow cytometry analysis. The specific gating strategies are listed in Supplementary Fig. 13.

**Knockout and rescue of *ssl7* and *ssl10* in *S. aureus* NCTC 8325**. To explore the effect of SSL10 on the cytotoxicity of *S. aureus*, *ssl10* or *ssl7* knockout strain was constructed using the vector pKOR1[51]. Briefly, ~1000 bp fragment upstream and downstream, respectively, of *ssl10* or *ssl7* was cloned to pKOR1 via lambda recombination (BP clonase enzyme mix, Invitrogen). The resulting plasmid was transferred via electroporation first to *S. aureus* RN4220 to modify DNA, and subsequently to *S. aureus* 8325. For homologous recombination and ingratiation of pKOR1 into the bacterial chromosome, *S. aureus* NCTC 8325 was grown at 43 °C on tryptic soy agar (TSA$_{Cm10}$), a non-permissive condition for pKOR1 replication. From the resulting plate, one colony was picked, inoculated into 1 ml TSB$_{Cm10}$ and incubated at 30 °C overnight to facilitate plasmid excision. Cultures were then spread on TSA containing 200 ng/ml anhydrotetracycline and incubated at 30 °C overnight for selecting *ssl10* or *ssl7* knockout *S. aureus* NCTC 8325. To rescue *ssl10* in *ssl10* knockout strain, the gene sequence of *ssl10* was ligated with *hprk* promoter and then cloned to the plasmid pOS1. The resulting plasmid was transferred via electroporation first to *S. aureus* RN4220 to modify DNA, and subsequently to *ssl10*-knockout *S. aureus* NCTC 8325. All strains were confirmed by Sanger sequencing (Sangon, Shanghai, China).

**Cytotoxicity assay of *S. aureus* *ssl10*-knockout or rescue strains**. WT, *ssl10* or *ssl7* knockout and rescue *S. aureus* NCTC 8325 strains were cultured in TSB medium (OXOID) for 8 h at 37 °C, 220 rpm. The same number of bacteria cells were inoculated into a new tube of TSB medium respectively, and cultured overnight at 37°C, 220 rpm for 8 h protein expression and secretion. Bacteria cells were collected after achieving stagnate phase and the supernatants were obtained by centrifugation at 5000 rpm for 15 min at room temperature. After filtration with a 0.22 μm filter, the supernatants were used to treat HEK293T or HUVEC cells for 48 h after 1:10 dilution with Opti-MEM reduced serum medium, and LDH released from the cells were determined. The level of LDH released was expressed as the fold of the control group (TSB medium-treated cells) after subtracting the background absorbance.

**Transmission electron microscopy**. HEK293T or HUVEC cells with or without SSL10 treatment were prefixed in Karnovsky's solution (1% paraformaldehyde, 2% glutaraldehyde, 2 mM calcium chloride, 0.1 M cacodylate buffer, pH 7.4) for 2 h and washed with cacodylate buffer. Postfixing was carried out in 1% osmium tetroxide and 1.5% potassium ferrocyanide for 1 h. After dehydration with 50–100% alcohol, the cells were embedded in Poly/Bed 812 resin (Pelco, Redding, CA, USA), polymerized and then observed under a HITACHI model H-7650 electron microscope (HITACHI, Tokyo, Japan).

**Immunoblotting**. The cells were lysed in RIPA buffer containing proteinase inhibitor (Sigma-Aldrich, St. Louis, MO, USA) and/or PhosSTOP Phosphatase Inhibitor (Roche, Basel, Switzerland) when necessary. A total of 30 μg of protein (determined by BCA protein quantification kit) for each sample was separated on 10% SDS-PAGE, transferred to a PVDF membrane, immunoblotted with appropriate antibodies as indicated. Antibody binding was detected using a luminescent image analyzer ImageQuant$^{TM}$ LAS 4000 mini (GE Healthcare Bio-Science AB, Uppsala, Sweden) after adding peroxidase-conjugated secondary antibodies and chemiluminescence substrates.

**CRISPR-Cas9 genome editing and rescue**. The E-CRISP online tool (http://www.e-crisp.org/E-CRISP/) was used to design specific single-guide RNAs targeting different genes. The 20-nucleotide guide sequence was annealed to the complementary oligos and then cloned into the pSpCas9 (BB)-2A-green fluorescent protein (GFP) plasmid (PX458; Addgene, Cambridge, MA, USA). HEK293T or HUVEC cells were transiently transfected with the CRISPR/Cas9 plasmids using Lipofectamine 3000 (Life Technologies, MA, USA) following the manufacturer's instructions. GFP-positive single clone was sorted using SmartSampler Analyzer (Beckman, California, USA) 48 h after transfection and cultured in a 96-well plate for about 10 days before confirming the gene knockout by sequencing and immunoblotting or flow cytometry. Sequences of the single-guide RNAs for various genes are listed in Table 1.

For RIPK3 rescue, RIPK3 cDNA was cloned into the pCDH-CMV-MCS-EF1-copGFP-T2A-Puro vector (Addgene, Cambridge, MA, USA). HEK293T WT and RIPK3 knockout cells were transiently transfected with the RIPK3-expression

plasmids using Lipofectamine 3000, and 48 h after transfection, SSL10-induced necroptosis was detected by LDH release assay.

**Mitochondrial membrane potential (ΔΨm) assay**. The mitochondrial potential, which reflects mitochondrial depolarization, was detected using the mitochondrial membrane potential assay kit following the manufacturer's instructions (Beyotime, Shanghai, China). JC-1 is a marker of mitochondrial activity. When the mitochondrial membrane potential (ΔΨm) is high, JC-1 will be present in the matrix of mitochondria as J-aggregate and emits red fluorescence. Conversely, when the ΔΨm is low, JC-1 is present as monomer and emits green fluorescence. The value of the JC-1 monomers to aggregates positive cells ratio quantifies the mitochondrial membrane depolarization. Briefly, JC-1 working solution was incubated with the cells in dark for 20 min at 37 °C. After three washes with prepared buffer, red and green fluorescence were detected on a flow cytometer using PI and FITC channels, respectively. The specific gating strategies are listed in Supplementary Fig. 13.

**Protein expression and purification**. DNA fragments encoding amino acid residues 31–227 of SSL10, 31–231 of SSL7, and SSL10 mutants were amplified by PCR from *S. aureus* strain Mu50 and cloned into the pET-22b (+) vector (Novagen) with a C-terminal $6 \times$ His-tag. WT and mutant SSL10 and SSL7 were expressed in *Escherichia coli* BL21 (DE3) and induced with 0.4 mM IPTG (isopropyl β-D-1-thiogalactopyranoside) for 4 h at 37 °C when $OD_{600}$ reached 0.6. The cells were harvested by centrifugation at 6000 rpm for 8 min, and lysed in a French press in lysis buffer [50 mM Tris-HCl, pH 7.5, 500 mM NaCl, 5% (v/v) Glycerol, 5 mM imidazole, 1 mM PMSF]. The lysate was centrifuged at 15,000 rpm for 30 min, and the supernatant was incubated with Ni-NTA resin for 30 min. The resin with target proteins were washed with 50 column volumes of washing buffer [50 mM Tris-HCl, pH 7.5, 500 mM NaCl, 5% (v/v) Glycerol, 40 mM imidazole] to remove contaminants and the target protein was eluted by elution buffer [50 mM Tris-HCl, pH 7.5, 500 mM NaCl, 5% (v/v) Glycerol, 300 mM imidazole]. The eluted protein was concentrated and further purified by Superdex 75 10/300 size exclusion column (GE Healthcare) equilibrated with the buffer containing 20 mM Tris-HCl, pH 7.5 and 200 mM NaCl. The entire protein purification procedure was carried out at 4 °C. Purity of the target protein was verified by SDS-PAGE and protein aliquots were stored at −80 °C for further use.

The DNA fragment of human TNFR1 extracellular domain (amino acid residues 22-211) was amplified by PCR using cDNA library of human spinal cord as template and cloned into the pET-28a (+) vector with an N-terminal MBP-tag, which was verified by Sanger sequencing. The recombinant protein MBP-TNFR1$^{ECD}$ and its CRD deletion mutants were expressed in *E. coli* Rosetta2 (DE3) strain and induced with 0.4 mM IPTG for 20 h at 16 °C when $OD_{600}$ reached 0.6, and were then purified by MBP-affinity chromatography.

**Crystallization**. SSL10 was concentrated to 4.3 mg/ml (in buffer containing 20 mM Tris-HCl, pH 7.5 and 200 mM NaCl), and used for initial crystallization trials by the sitting-drop vapor-diffusion method at 16 °C with index crystallization screen kit. Crystals were obtained from the buffer containing 2.1 M DL-malic acid, pH 7.0.

**Data collection, structure determination and refinement**. The crystals of SSL10 were soaked in cryoprotectant buffer consisting 2.1 M DL-Malic acid, pH 7.0 and 20% Glycerol for several seconds and flash-cooled in liquid nitrogen. X-ray diffraction data was collected at beamline BL18U1 of Shanghai Synchrotron Radiation Facility. Diffraction data were processed, integrated, and scaled using HKL2000[52].

The crystal structure of SSL10 was determined by molecular replacement using the program Phaser in the CCP4i suite[53,54] with Exotoxin SACOL0473 (PDB code 3R2I) as the search model. After several runs of structure refinement using the programs REFMAC5, Phenix and Coot[55–57], the final model was refined to 1.9 Å resolution with $R_{work}$ of 20.94% and $R_{free}$ of 24.62%. Data collection and structure refinement statistics are summarized in Table 2. All figures of protein structure were prepared using PyMOL (http://www.pymol.org).

**Docking analysis of SSL10 and TNFR1$^{ECD}$**. SSL10-TNFR1$^{ECD}$ model was generated using the HawkDock webserver (http://cadd.zju.edu.cn/hawkdock/)[34] with the structure of SSL10 and the structure of extracellular domain of TNFR1 (PDB code: 1EXT). The top 10 predicted models obtained were re-ranked by MM/GBSA method for binding interfaces analysis[35].

**MBP pull-down assay**. Thirty microgram of MBP-TNFR1$^{ECD}$ was incubated with 30 μg of WT or mutant SSL10 for 1 h on ice, and the protein mixture was centrifuged at 15,000 rpm for 30 min at 4 °C to remove precipitates. The supernatant was then added into 1 ml binding buffer (20 mM Tris-HCl, pH 7.5, 200 mM NaCl, 0.5% NP-40) with 20 μl MBP beads, and incubated for 1 h at 4 °C. The beads were washed with 1 ml binding buffer for four times to remove any non-specific bindings, and the proteins bound to MBP resin were eluted by 20 μl elution buffer (20 mM Tris-HCl, pH 7.5, 200 mM NaCl, 50 mM D-Maltose) and analyzed by 15% SDS-PAGE.

**Isothermal titration calorimetry (ITC)**. The interaction between SSL10 protein and MBP-TNFR1$^{ECD}$ was analyzed by ITC by using a MicroCal PEAQ-ITC instrument (Malvern) at 16 °C. Sixty microliter SSL10 or its variants (500–600 μM) were injected into a sample cell containing 280 μl MBP, MBP-TNFR1$^{ECD}$ or its mutants (~60 μM) in binding buffer (20 mM Tris-HCl, pH 7.5, 200 mM NaCl). Titration data were analyzed by the MicroCal PEAQ-ITC Analysis Software (Malvern) using one set of sites fitting model.

**Statistics and reproducibility**. Statistical analyses were performed with GraphPad Prism Version 6.0 (GraphPad Inc., La Jolla, CA, USA) software program. The comparisons between two groups were analyzed by one-way or two-way ANOVA. All error bars in the graphs show standard deviation (±SD). $p$ values <0.05 were considered statistically significant. Sample replicates are described in the corresponding legends.

**Multiple sequence alignment**. Protein sequences used for alignment were obtained from the Uniprot database (https://www.uniprot.org/). The accession number of SSL3, SSL7, SSL8, SSL10, and SSL11 are A0A0H3JPZ9, A0A0H3JTD4, A0A0H3JQE0, A0A0H3JQ04, and A0A0H3JXK4, respectively. Multiple sequence alignment was performed using the MultAlin software (http://multalin.toulouse.inra.fr/multalin/) and ESPript software (https://espript.ibcp.fr/ESPript/ESPript/). Sequence identities were calculated by Clustal Omega (https://www.ebi.ac.uk/Tools/msa/clustalo/).

**Reporting summary**. Further information on research design is available in the Nature Research Reporting Summary linked to this article.

## Data availability
Structural data of TNFR1$^{ECD}$ used for docking analysis are available with the PDB code 1EXT. All raw data underlying the graphs and charts presented in the main and Supplementary Figures are present in Supplementary Data 1. Unedited gel images are included in Supplementary Fig. 14. All other data are available from the corresponding authors on the reasonable request.

## Code availability
The structure data of SSL10 generated in this study have been deposited to the Protein Data Bank (PDB, https://www.rcsb.org) under the following accession number: PDB code 6LWT.

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

## Acknowledgements

The authors would like to thank the staff at beamline BL18U1 of the Shanghai Synchrotron Radiation Facility of the National Facility for Protein Science in Shanghai for the assistance with data collection. This work was supported by the National Natural Science Foundation of China (81801974 to J.H., 91853133 to J.Z., 81701936 to Y.T., and 81971890 to X.M.), the Natural Science Foundation of Fujian Province of China (2020J01615), the National Key Research and Development Program of China (2017YFA0503600, 2016YFA0400903), the Foundation for Innovative Research Groups of the National Natural Science Foundation of China (31621002), Shanghai Key Laboratory of Clinical Molecular Diagnostics for Pediatrics (20dz2260900), Shanghai Key Laboratory of Emergency Prevention, Diagnosis and Treatment of Respiratory Infectious Diseases (20dz2261100), the Innovative Program of Development Foundation of Hefei Center for Physical Science and Technology (2017FXCX004), Users with Excellence Project of Hefei Science Center CAS (2018HSC-UE001), China Postdoctoral Science Foundation (2017M621492), Joint Funds for the innovation of science and Technology, Fujian province (Grant number: 2020Y9006), and USTC Research Funds of the Double First-Class Initiative.

## Author contributions

N.J., Y.T., J.Z., X.M., and J.H. designed the experiment and wrote the paper; N.J., G.L., L.C., X.Z., and J.H. performed the experiments; X.W. provided the Staphylococcal aureus strains obtained from blood culture of the patients; Q.C. and XL.M. evaluated the clinical diagnosis of hypoproteinemia, septic shock, and multiple organ dysfunction syndrome; W.C., C.W., and X.Z. generated the crystal structure of SSL10; Y.T., J.Z., X.M., and J.H. supervised the research. All authors participated in the data analysis.

## Competing interests

The authors declare no competing interests.
