## [Peer Review File · Communications Biology]

Reviewers' comments:

Reviewer #1 (Remarks to the Author):

This manuscript describes a study of *S. aureus* SSL10 induced necroptosis of mammalian cells through interaction with a TNF receptor and activation of the RIPK3 dependent signal pathways. It is well written and several different experimental approaches are used to support the authors' conclusions. Of greatest concern is the biological significance of the SSL10 induced necroptosis owing to the long period of incubation necessary to obtain a biological effect compared to the activity of other *S. aureus* cytotoxins. The 48 hours required to see activity may also affect the interpretation of the experimental results.

Specific comments and questions:

1. Line 80, reference 7 (Kitur et al) states that *S. aureus* supernatant and also specific virulence proteins induce necroptosis within 2 hours. SSL10 takes 48 hours to cause this effect. In fact, figure S1 show there are more viable cells at day 2 following treatment than day 0. How can you reconcile the data from figure S1 with figure 1B that shows a significant decrease in viability with the same concentration of SSL10 on day 2? A case needs to be made for the biological relevance of SSL10.
2. Line 124, Figure 1 C This and similar figures are key to proving that SSL10 does not induce apoptosis by showing minimal display of annexin V. However, the only comparison of SSL10 treatment is with cells harvested at 48 hours, they have a broad range of annexin reactivity and the PI positive SSL10 treated cells seem to have a higher overall FITC fluorescence than untreated cells. How can we be certain that the untreated cells are not undergoing apoptosis and that the gate should be placed further to the left, which would indicate apoptosis has occurred? These experiments would benefit from showing the profile of cells prior to incubation and a positive control for apoptosis.
3. Figure 1, 2, 3, and 4. The labeling of the vertical axis is inconsistent. The term "fold of SSL10 treated cells" should indicate that these cells have a value of 1. In figures 1 and 2 they do, in figures 3 and 4 more often the value of untreated cells is set to 1 but the label is not changed.
4. Figure 3 G. The PE and FITC labels are very difficult to read and may be incorrect. Are you actually detecting PE and FITC or just using these excitation and emissions conditions for the JC-1 experiment?
5. Line 192. The amino acid numbers need to be referenced to a sequence or Figure 6C. You should also indicate where this sequence came from and how well it is conserved among *S. aureus* isolates.

Reviewer #2 (Remarks to the Author):

Brief summary and overall impression of the manuscript: The authors showed new physiological of SSL10 that shows cytotoxicity via TNFR1-RIPK3 pathway. The finding provides a novel function of SSL proteins.

Specific comments and recommendation:

1. In this manuscript, the authors show the release of LDH from SSL10 treated cell by the fold of the release in compare with control cells. The author should show the amount of LDH release from control cells in each data and show the percentage of LDH release in compare with total LDH content of the cells.
2. In Figure 1F, the author shows the supernatant of *S. aureus* exhibits cytotoxicity to HEK293 and HUVEC cells. To clarify the contribution of SSL10 in the supernatant, the authors should show the amount of SSL10 in the supernatant.

Reviewer #3 (Remarks to the Author):

The manuscript by Jia et al. identifies necroptotic properties, and characterises the mechanism of

action, of a *S. aureus* virulence factor called SSL-10. The authors demonstrate that SSL-10 induces necroptosis of HEK and HUVEC cells and indicate this occurs through engagement of TNFR1, using cell based assays. The crystal structure of SSL-10 was solved by X-ray diffraction and a putative interface between SSL-10 and TNFR1 was modelled and assessed through site-directed analysis.

Overall, the findings are of interest for uncovering a pathway in which *S. aureus* can induce necroptosis of host cells. However, the characterisation of the SSL-10 and TNFR1 interaction is weak and not well supported. Additional controls are required in many cell based assays given that LPS endotoxins have not been removed from protein purifications. The structural detailing of SSL-10 is an important step in addressing how this molecule interacts with its target ligands and is novel in itself. However, SSL10-ligand interactions are poorly assessed in its current form.

Major comments

1) The cell based assays indicate that SSL-10 induces necroptosis of cells through TNFR1 - however I think further assays are required to support this interaction. anti-TNFR1 blocking mAb could be utilised to block binding of SSL10-GFP (Fig S3) or used to block necroptosis on HUVEC cells.

There is only very weak biochemical data presented that supports a protein-protein interaction between SSL-10 and TNFR1. In Fig 4 a pull-down of SSL-10 by TNFR1 would support an interaction; however additional biochemical analyses are required to confirm the specificity and the affinity of the interaction. I suggest that ELISAs, Western Blots, isothermal calorimetry and SPR approaches are employed to confirm the specificity and affinity of the SSL-10 and TNFR1 interaction.

2) The authors correctly state in the introduction that SSL-10 interacts with other components of the human immune system - the CXCR4 receptor, IgG, prothrombin and factor Xa. However, the SSL-10 and TNFR1 interaction is not studied or discussed in the broader context of SSL-10 biology. For example, do these molecules compete with each other to interact with SSL-10 (interactions could be modelled, or competitive inhibition assays)? which molecule target has the highest affinity for SSL-10? how does SSL-10 impact immune responses through these combined interactions?

3) This study provides the first structural information for SSL-10 molecule. Given that SSL-10 is structurally related to the remaining 13 SSL molecules, it would be beneficial to compare the structure of SSL-10 against the characterised structures of other SSLs (not just SSL-7). For example, structural information has been resolved and is available for SSL-11 (PDB: 2RDH), SSL-5 (set-3, PDB: 1M4V), SSL-4 (PDB: 4DXG). How does variation in SSL-10 vs other SSLs account for differences in binding TNFR1 and other ligands?

4) analysis of the SSL-10 and TNFR1 interface is weak. Firstly, the analysis of the predicted structure of SSL-10 and TNFR1 interaction through HDOCK analysis is incomplete in its current form as there is no methodology presented. How many iterations of the docking analysis were performed? What was the mean docking score and mean ligand rmsd? what proportion of the top 10, 20, 50 hits was each of the residues listed in the results identified as being located at the interface? and what were the scores associated with those predictions? Additional approaches could be beneficial such as hawkdock that can calculate free binding energy of individual residues in docked molecules. Secondly, the analysis showing that the mutated forms (A or B) of SSL-10 do not interact with TNFR1 is very weak. Analysis in Fig 6D indicates mutation does not abolish necroptotic activity of SSL-10. Analysis in Fig 6E and 6F shows reduced pulldown of SSL-10 mutants compared to SSL-10 wildtype by TNFR1. Neither dataset indicate that the mutated SSL-10 residues are critical for TNFR1 interaction, otherwise signals would be 0. Thus the conclusion "Therefore, the docking and mutagenesis analyses demonstrate that 259 residues H64, K66, N85, S88, Q91, K206, K208, and Y209 are critical for SSL10 binding with TNFR1ECD 260 and subsequent initiation of the necroptosis signal cascade." is not supported. Complete in silico analysis and further biochemical analysis is required to support this conclusion. Furthermore, what are the critical residues in TNFR1 required for SSL-10 interaction? what support is there for this interaction?

5) SSL-10 is purified from *E. coli* cultures. LPS endotoxins are well known to be contaminants of protein purifications and have capacity to induce necroptosis. Given the nature of the study, it is critical that LPS endotoxins are removed from SSL-10 and other protein purifications and that protein controls are used throughout the cell-based assays. What strategies have been used to eliminate LPS from protein purifications? how has this been validated? Protein controls (such as SSL7) should be used throughout the cell based assays in Fig 1 to 5.

Minor comments

- 1) what is the predicted interface of SSL-10 for other ligands? has this been characterised in previous studies? what does in silico analysis show?
- 2) line 62 and 77 - given that much of the manuscript identifies a new *S. aureus* molecule that induces necroptosis, more background information is required that describes which *S. aureus* molecules are currently known to induce necroptosis and their mechanisms of action.
- 3) line 86 - change to staphylococcal superantigens?
- 4) lines 87-88 is inaccurate. The 14 ssl genes are encoded at two different loci - not 1 pathogenicity island. Do all strains have genes encoding all SSL molecules? Is the gene encoding SSL-10 present in all strains?
- 5) line 97 - what makes SSL10 "novel" compared to other SSLs? I thought it was one of the best characterised SSLs, and therefore probably the least novel?
- 6) lines 119-120. The design of the study is not well described. What was the justification for studying ssl3, 7, 8, 10 and 11 only?
- 7) Fig 1A and B - there is no protein control in these assays. Surely an alternative SSL could be used as a better control than buffer?
- 8) lines 130-132. There is no data presented that demonstrates SSL10 is present in the *S. aureus* culture supernatants - or the quantity. How do you know SSL10 is expressed by 8235-4 and released into the supernatant?
- 9) Fig2A and 2B - there looks to be a marginal if at all a difference between Nec1 and GSK treatments. the results here are overstated
- 10) lines 183-184. which other receptors induce necroptosis? what was the justification for studying TNFR1 as opposed to any other receptor?
- 11) why does TNFR1 knockdown only partly suppress SSL10 induced necroptosis? Fig 4C
- 12) data presentation in Fig 4 is a bit backwards to me. It would be best to show reduced binding of SSL10-GFP to knockdown cells (F), then reduced LDH release (B), increased cell viability (D, E)
- 13) Fig 6A. The individual residues to be highlighted cannot be seen. They should be presented in an alternative colour to the remainder of the chain.
- 14) Fig 6B. Residues of interest cannot be visualised.
- 15) Fig 6C. SSL-7 and SSL-10 are both variable. The sequences in the alignment are from which genome? Do any SSL10 variants have mutations are the residues identified? Do any SSL7 variants have the residues identified for TNFR1 interaction?

Point-to-point Response Letter

Please find our responses to the specific comments raised by the reviewers below. We have copied each comment in *Italic*, which is followed by our point-by-point response in blue. Proper changes have been incorporated into the revised manuscript with all changes being tracked.

Reviewer #1: (Remarks to the Author)

*This manuscript describes a study of *S. aureus* SSL10 induced necroptosis of mammalian cells through interaction with a TNF receptor and activation of the RIPK3 dependent signal pathways. It is well written and several different experimental approaches are used to support the authors' conclusions. Of greatest concern is the biological significance of the SSL10 induced necroptosis owing to the long period of incubation necessary to obtain a biological effect compared to the activity of other *S. aureus* cytotoxins. The 48 hours required to see activity may also affect the interpretation of the experimental results.*

Thank you for taking your time to review our paper.

Specific comments and questions:

*1. Line 80, reference 7 (Kitur et al) states that *S. aureus* supernatant and also specific virulence proteins induce necroptosis within 2 hours. SSL10 takes 48 hours to cause this effect.*

Response: Thanks for the question. We do agree that several *S. aureus* pore-forming toxins, including α -toxin, PSM and PVL, have been reported to induce necroptosis within 2 h^{1,2}. However, in different systems, the time to induce necroptosis is different. For example, HT-29 cells have been shown to undergo necroptosis 8 h after treatment with TNF α , Smac-mimetic and Z-VAD (TSZ)³, while Z-VAD and 5-FU take 48 h to trigger necroptosis in HT-29 cells⁴. *Salmonella enterica* serovar Typhimurium has been shown to induce necroptosis of macrophages at 24 h or 48 h after infection, with no death in 6 h⁵. Therefore, it is reasonable to speculate that the time for necroptosis induction may be dependent on the type of toxic proteins produced by *S. aureus* and

different types of cell model used in the experiments.

In fact, figure S1 show there are more viable cells at day 2 following treatment than day 0. How can you reconcile the data from figure S1 with figure 1B that shows a significant decrease in viability with the same concentration of SSL10 on day 2?

Response: We are sorry about the confusing data presentation. In Fig. 1b, the vertical axis shows the fold of SSL10-treated cells on day 2 compared to untreated cells on day 2. However, in Fig. S1, the vertical axis shows the fold of SSL10-treated cells on day 2 compared to untreated cells on day 0. We have reanalyzed the viability of SSL10-treated cells on day 2 in Fig. S1, which is normalized to that of untreated cells on day 2, and the data is similar to that from Fig. 1 in the revised manuscript (Fig. S1).

Fig. S1. Cytotoxicity of different SSL proteins on HUVEC. Cell viability of HUVEC cells treated with 2 μM SSL3, SSL7, SSL8, SSL10, or SSL11 for 24 h or 48 h was detected by MTS assay, and was then normalized to that of the untreated cells (*i.e.*, Ctrl cells, buffer treated cells) on the same day. All data represent mean ± SD from at least three independent experiments. *, $p < 0.05$ compared to the Ctrl cells (buffer-treated cells).

A case needs to be made for the biological relevance of SSL10.

Response: Thanks for your suggestion. Reijer *et al.* characterized the serial levels of IgG and IgA antibodies against 56 staphylococcal antigens in multiple serum samples of 21 patients with *S. aureus* bacteremia. Their data showed that an increase in IgG

levels against SSL10 was observed at some time points after the onset of bacteremia in 95 to 100% of all patients⁶. In addition, we observed that mRNA levels of *ssl10* in the strains isolated from the patients with hypoproteinemia, septic shock or multiple organ dysfunction syndrome (MODS) were significantly higher than that in strains from patients without these diagnoses, whereas the mRNA levels of *ssl7* in these strains were similar (Fig. S10). Since endothelial damage is a hallmark of hypoproteinemia, septic shock and MODS, these data suggest that SSL10-induced necroptosis of endothelial cells might be correlated with the progression of *S. aureus* bacteremia. We have included the information in the discussion of the revised manuscript.

Fig. S10 mRNA levels of *ssl10* and *ssl7* in *S. aureus* strains. The gene expression levels of *ssl10* (a) and *ssl7* (b) isolated from the blood cultures of sixteen children infected by *S. aureus* were determined by real-time qPCR. The data were analyzed by the nonparametric Mann-Whitney test using GraphPad Prism Version 6.0 software program. *, $p < 0.05$; n.s., not significant.

2. Line 124, Figure 1 C This and similar figures are key to proving that SSL10 does not induce apoptosis by showing minimal display of annexin V. However, the only comparison of SSL10 treatment is with cells harvested at 48 hours, they have a broad

range of annexin reactivity and the PI positive SSL10 treated cells seem to have a higher overall FITC fluorescence than untreated cells. How can we be certain that the untreated cells are not undergoing apoptosis and that the gate should be placed further to the left, which would indicate apoptosis has occurred? These experiments would benefit from showing the profile of cells prior to incubation and a positive control for apoptosis.

Response: Thank you for the suggestion. As suggested, we showed the profile of cells prior to incubation, SSL7 treatment for 2 days, and a positive control (cisplatin) for apoptosis. Our results showed that without treatment, the cell viability was similar on day 0 and day 2. In addition, under the same gating criteria, the patterns of SSL10-treated cells were different from cisplatin-treated cells, which are known to undergo apoptosis. AnnexinV/PI detection indicated a much higher AnnexinV⁺PI⁻ ratio, representing early apoptotic cells, in the group challenged with cisplatin, but not with SSL10 (Fig. S2). These data indicate that SSL10 triggers cell necrosis rather than apoptosis.

Fig. S2. SSL10-treated cells undergo necrosis. HUVEC were treated with buffer (Ctrl), SSL7 or SSL10 for 48 h. HUVEC before treatment and HUVEC treated with cisplatin for 24 h were used

as a negative control and a positive control for cellular apoptosis, respectively. All cells were detected by flow cytometry after Annexin V/PI staining. The dot plot (a) is representative of three independent experiments, and the quantification results are shown as a bar graph (b). All data represent mean \pm SD from at least three independent experiments. *, $p < 0.05$, **, $p < 0.01$; n.s., not significant as indicated.

3. Figure 1, 2, 3, and 4. The labeling of the vertical axis is inconsistent. The term “fold of SSL10 treated cells” should indicate that these cells have a value of 1. In figures 1 and 2 they do, in figures 3 and 4 more often the value of untreated cells is set to 1 but the label is not changed.

Response: Sorry about the confusing data presentation. We have changed the vertical axis “fold of -SSL10-treated cells” to “fold of untreated cells” in Fig. 1a, with the value of untreated cells set to 1. In other relevant figures, we have changed the vertical axis to show the percentage of LDH release to the medium compared to the total LDH level of the cells (*i.e.*, % of LDH released) according to other reviewer’s comments.

4. Figure 3 G. The PE and FITC labels are very difficult to read and may be incorrect. Are you actually detecting PE and FITC or just using these excitation and emissions conditions for the JC-1 experiment?

Response: Sorry for the unclear presentation. We just used the excitation and emission conditions of PI and FITC channels for the JC-1 experiment. JC-1 is a marker of mitochondrial activity, and when the mitochondrial membrane potential ($\Delta\Psi_m$) is high, JC-1 will be present in the matrix of mitochondria as aggregate and emits red fluorescence (the PI channel). Conversely, when the $\Delta\Psi_m$ is low, JC-1 is present as monomer and emits green fluorescence (the FITC channel). We have labeled the x and y axis more clearly in the revised figure (Fig. 3e and Fig. 4d. Fig. 3e is shown below as an example), and the detailed information has been added in Methods

section (Line 484-489) in the revised manuscript.

e

Fig. 3e. Depolarization of the mitochondrial membrane of wild type (WT) HEK293T cells or RIPK3 knockout cells treated by SSL10 was measured by flow cytometry after JC-1 staining. The dot plot (left) is representative of three independent experiments, and the results of quantification are shown as a bar graph (right).

5. Line 192. The amino acid numbers need to be referenced to a sequence or Figure 6C. You should also indicate where this sequence came from and how well it is conserved among *S. aureus* isolates.

Response: Thank you for the suggestion. In the sentence “TNFR1^{ECD} (the extracellular domain of TNFR1 containing amino acids 22-211)”, the amino acid numbers are referred to the sequence of human TNFR1 (GenBank number: CAA39021.1). In order to avoid any confusion, we have changed the sentence to “TNFR1^{ECD}, the extracellular domain of human TNFR1 containing amino acids 22-211, (GenBank number: CAA39021.1)” in the revised manuscript (Line 205). The sequence of SSL10 used in the present study came from *S. aureus* Mu50, which is the same as NCTC 8325 (this has been indicated in Line 262 in the revised manuscript). Sequence alignment shows that SSL10 is highly conserved in *S. aureus* (Fig. R1).

Strains	NCTC8325	N315	NCTC10654	Newman	USA300	MRSA252	NCTC7878	MN8
Sequence identity to SSL10 from Mu50	100%	100%	100%	100%	99.56%	82.82%	82.82%	82.82%

Fig. R1 Sequence alignment of SSL10 from different *S. aureus* strains. Sequence alignment of SSL10 from *S. aureus* strains Mu50, NCTC8325, N315, NCTC10654, Newman, USA300, MRSA252, NCTC7878 and MN8 was conducted with MultAlin and ESPrInt. Sequence identities of SSL10 from Mu50 to SSL10 from other strains calculated by Clustal Omega are listed below.

Reviewer #2 (Remarks to the Author):

Brief summary and overall impression of the manuscript: The authors showed new physiological of SSL10 that shows cytotoxicity via TNFR1-RIPK3 pathway. The finding provides a novel function of SSL proteins.

Thank you for your positive answer and taking your time to review our paper.

1. In this manuscript, the authors show the release of LDH from SSL10 treated cell by the fold of the release in compare with control cells. The author should show the amount of LDH release from control cells in each data and show the percentage of LDH release in compare with total LDH content of the cells.

Response: Thank you for the suggestion. We quantitatively measured lactate dehydrogenase (LDH) release from cells in each group by the CytoTox 96® Assay (Promega, G1780), and according to the absorbance at 490 nm, the percentage of

LDH release in comparison with total LDH content of the cells was shown as suggested in all relevant figures in the revised manuscript. Fig. 1e was shown below as an example.

Fig. 1e. Cells were pretreated with 10 μ M Z-VAD-fmk for 30 min at 37 $^{\circ}$ C and 5% CO₂, and then stimulated with 2 μ M SSL10. The release of LDH was detected. **, $p < 0.01$; ***, $p < 0.001$ compared to the Ctrl cells (buffer-treated cells). n.s., not significant compared to the SSL10-treated cells.

2 In Figure 1F, the author shows the supernatant of *S. aureus* exhibits cytotoxicity to HEK293 and HUVEC cells. To clarify the contribution of SSL10 in the supernatant, the authors should show the amount of SSL10 in the supernatant.

Response: Thanks for your suggestion. Since commercial antibodies for SSL10 are unavailable, we constructed an 8325-EGFP strain in which the gene *egfp* was fused to the C-terminal of *ssl10* (Fig. R2a). The presence of SSL10-EGFP in the supernatant can be clearly detected by anti-GFP antibody (Fig. R2b).

Fig. R2. (a) Genomic schematic diagram of NCTC 8325 and NCTC 8325-EGFP strain. (b) SSL10-EGFP detection in the supernatant of NCTC 8325 and 8325-EGFP strains with anti-GFP antibody by immunoblotting.

Reviewer #3 (Remarks to the Author):

The manuscript by Jia et al. identifies necroptotic properties, and characterises the mechanism of action, of a S. aureus virulence factor called SSL-10. The authors demonstrate that SSL-10 induces necroptosis of HEK and HUVEC cells and indicate this occurs through engagement of TNFR1, using cell based assays. The crystal structure of SSL-10 was solved by X-ray diffraction and a putative interface between SSL-10 and TNFR1 was modelled and assessed through site-directed analysis.

Overall, the findings are of interest for uncovering a pathway in which S. aureus can induce necroptosis of host cells. However, the characterisation of the SSL-10 and TNFR1 interaction is weak and not well supported. Additional controls are required in many cell based assays given that LPS endotoxins have not been removed from protein purifications. The structural detailing of SSL-10 is an important step in addressing how this molecule interacts with its target ligands and is novel in itself. However, SSL10-ligand interactions are poorly assessed in its current form.

Thank you for taking your time to review our paper.

1) The cell based assays indicate that SSL-10 induces necroptosis of cells through TNFR1 - however I think further assays are required to support this interaction. anti-TNFR1 blocking mAb could be utilised to block binding of SSL10-GFP (Fig S3) or used to block necroptosis on HUVEC cells.

Response: Thank you for the suggestion. As shown in Fig. S5a, anti-TNFR1 mAb impaired SSL10-induced necroptosis of HUVEC in a dose-dependent manner. Additionally, we also observed a dose-dependent inhibitory effect of purified MBP-tagged TNFR1 extracellular domain (TNFR1^{ECD}) against the cytotoxic effect of SSL10 (Fig. S5b). These data, along with the results presented in Fig. 4, provided strong evidence that SSL10 induces cell necroptosis mainly through TNFR1.

Fig. S5 Competitive inhibition of SSL10-induced cytotoxicity. HUVEC cells were pretreated with anti-TNFR1 antibody (a) or purified MBP-TNFR1^{ECD} (b) at different doses, and then treated with 2 μM SSL10 as indicated. The LDH release was detected. Cells treated with 2 μM SSL7 were used as a negative protein control throughout the experiments. All data represent mean ± SD from at least three independent experiments. **, $p < 0.01$; ***, $p < 0.001$ compared to the Ctrl cells (buffer-treated cells). #, $p < 0.05$; ##, $p < 0.01$ compared to the SSL10-treated cells.

2) There is only very weak biochemical data presented that supports a protein-protein interaction between SSL-10 and TNFR1. In Fig 4 a pull-down of SSL-10 by TNFR1 would support an interaction; however additional biochemical analyses are required to confirm the specificity and the affinity of the interaction. I suggest that ELISAs, Western Blots, isothermal calorimetry and SPR approaches are employed to confirm

the specificity and affinity of the SSL-10 and TNFR1 interaction.

Response: Thanks for your suggestion. As suggested, we have performed immunoblotting and isothermal titration calorimetry (ITC) assays to confirm the specificity and affinity of the interaction between SSL10 and TNFR1. Immunoblotting results demonstrated that purified TNFR1^{ECD} could specifically interact with SSL10, but not with SSL7 (Fig. 4e). ITC assays showed that the K_D value between TNFR1^{ECD} and SSL10 is $3.87 \pm 0.11 \mu\text{M}$, but no binding was detected between SSL7 and TNFR1^{ECD} (Fig. 4f). A previous report showed that surface receptors typically have a medium binding affinity to their ligands, with K_D value in the range of 10^{-6} ~ 10^{-10} M⁷. Therefore, the binding affinity of SSL10 to the extracellular domain of TNFR1 is comparable to that of other surface receptors to their ligands.

Fig. 4e and f. (e) Immunoblotting of SSL10 after pull-down with MBP-TNFR1^{ECD}. (f) ITC assays for SSL10 binding to MBP-TNFR1^{ECD} or MBP control protein.

3) *The authors correctly state in the introduction that SSL-10 interacts with other components of the human immune system - the CXCR4 receptor, IgG, prothrombin and factor Xa. However, the SSL-10 and TNFR1 interaction is not studied or discussed in the broader context of SSL-10 biology. For example, do these molecules compete with each other to interact with SSL-10 (interactions could be modelled, or competitive inhibition assays)? which molecule target has the highest affinity for SSL-10? how does SSL-10 impact immune responses through these combined interactions?*

Response: Thanks for the questions. It has been reported that SSL10 can interact with

other components of the human immune system - such as the CXCR4 receptor, IgG, prothrombin and factor Xa. The binding sites on SSL10 for these different targets are different. The binding site for IgG1 predominantly resides in the N-terminal OB-fold domain of SSL10⁸, while the prothrombin binding regions are distributed on both the N- and C-terminal domains of SSL10: β 1- β 4 in OB-fold domain and β 13- β 15 in β -grasp domain, corresponding to sequence 54-81 (MEM KN ISALK HGKNN LRFKF RGIKI QVL) and 195-227 (SFYNL DLRSK LKFKY MGEVI ESKQI KDIE VNLK), respectively⁹. So far, the binding sites on SSL10 for CXCR4 and Factor Xa have not been determined. In our study, residues important for TNFR1 binding are H64, K66, N85, S88, Q91, K206, K208 and Y209, which are also distributed on both two domains of SSL10: the N-terminal OB-fold domain and the C-terminal β -grasp domain. Therefore, this separate binding domain of SSL10 to TNFR1 could potentially allow for simultaneous binding of several other proteins in a similar way to SSL7, which simultaneously binds to complement C5 and IgA¹⁰. It should also be noted that when the binding sites of other proteins on SSL10 are proximal to or overlap with the TNFR1 binding sites, these molecules could compete with each other to interact with SSL10. For instance, the potential binding sites on SSL10 for the prothrombin overlap with TNFR1 binding sites, and therefore the λ -carboxyglutamic acid (Gla) domain of prothrombin, which is essential for its interaction with SSL10¹¹, could competitively inhibit the cytotoxicity of SSL10 in a dose-dependent manner (Fig. S5c), although the inhibitory effect of purified Gla domain of prothrombin was not as strong as TNFR1^{ECD} (Fig. S5b). These points has been added in the discussion section (Line 307-321 in the revised manuscript).

Fig. S5 Competitive inhibition of SSL10-induced cytotoxicity. HUVEC cells were pretreated with purified MBP-TNFR1^{ECD} (b) or GST-Gla (c) at different doses, and then treated with 2 μM SSL10 as indicated. The LDH release was detected. Cells treated with 2 μM SSL7 were used as a negative protein control throughout the experiments. All data represent mean ± SD from at least three independent experiments. **, $p < 0.01$ compared to the Ctrl cells (buffer-treated cells). #, $p < 0.05$; ##, $p < 0.01$ compared to the SSL10-treated cells.

The binding affinity of SSL10 to IgG1 and prothrombin are comparable with each other, with the K_D values of 0.32 μM and 0.136 μM, respectively^{8,11}. The K_D value of SSL10 binding to TNFR1 is 3.87 ± 0.11 μM, which is lower than that to IgG1 or prothrombin if we don't count the effects caused by different experimental conditions. The binding affinity of SSL10 to other ligands such as CXCR4 has not been studied previously thus cannot be compared.

SSL10 shows strong sequence conservation in *S. aureus*. Previous studies demonstrate that SSL10 facilitates immune escape by interacting with CXCR4, IgG1 and prothrombin. Here, we found that SSL10 could induce necroptosis. Because IgG1 and prothrombin may have higher binding affinity with SSL10 than TNFR1, we speculated that SSL10 may play different roles during *S. aureus* infection. In the early stage of infection during which the SSL10 concentration may be relatively low, SSL10 could contribute to immune escape by interacting with IgG1 and prothrombin,

maybe with CXCR4 as well. In the late stage during which the SSL10 concentration is higher, SSL10 might be involved in the invasion and diffusion of *S. aureus* through killing vascular endothelial cell by interacting with TNFR1.

4) This study provides the first structural information for SSL-10 molecule. Given that SSL-10 is structurally related to the remaining 13 SSL molecules, it would be beneficial to compare the structure of SSL-10 against the characterised structures of other SSLs (not just SSL-7). For example, structural information has been resolved and is available for SSL-11 (PDB: 2RDH), SSL-5 (set-3, PDB: 1M4V), SSL-4 (PDB: 4DXG). How does variation in SSL-10 vs other SSLs account for differences in binding TNFR1 and other ligands?

Response: Thanks for your suggestions. We performed structural alignment of SSL10 with SSL3ΔN (PDB: 5D3D), SSL4 (PDB: 4DXG), SSL5 (PDB: 1M4V), SSL7 (PDB: 3KLS), SSL8 (PDB: 3R2T) and SSL11 (PDB: 2RDH), and the RMSD values were 0.903 Å, 0.939 Å, 1.013 Å, 1.345 Å, 0.823 Å, and 0.821 Å, respectively. As shown in Fig. S7b, the overall structures of these SSL proteins are similar, while major differences are located in the conformations of the flexible loops and in the linker between the N- and C- terminal domains. As shown in Fig. S7c, SSL10 has a large positively charged surface, which is more than that in other SSL proteins. This may contribute to the binding of SSL10 to TNFR1, as the extracellular domain of TNFR1 has more negative and neutral surface charge (Fig. R3).

Fig. S7. Structure comparison of SSL10 with other SSL proteins. (b) Superposition of the structure of SSL10 onto that of SSL3ΔN, SSL4, SSL5, SSL7, SSL8, and SSL11, respectively. SSL10 is colored cyan, while SSL3ΔN, SSL4, SSL5, SSL7, SSL8, and SSL11 are colored gray, deepsalmon, yellow, orange, pink, and purple, respectively. (c) The electrostatic surface potentials of SSL10, SSL3ΔN, SSL4, SSL5, SSL7, SSL8, and SSL11. Positively and negatively charged surface regions are colored blue and red, respectively.

Fig. R3. The electrostatic surface potentials of TNFR1^{ECD} (PDB: 1EXT). Positively and negatively charged surface regions are colored blue and red, respectively.

5) *Analysis of the SSL-10 and TNFR1 interface is weak. Firstly, the analysis of the predicted structure of SSL-10 and TNFR1 interaction through HDOCK analysis is incomplete in its current form as there is no methodology presented. How many iterations of the docking analysis were performed? What was the mean docking score and mean ligand rmsd? what proportion of the top 10, 20, 50 hits was each of the residues listed in the results identified as being located at the interface? and what were the scores associated with those predictions? Additional approaches could be beneficial such as hawkdock that can calculate free binding energy of individual residues in docked molecules.*

Response: Thanks for the suggestion. We have performed docking analysis to obtain SSL10-TNFR1 interaction model from the HDOCK webserver. One round of docking analysis was performed and 10 different docking models were obtained. The mean docking score and mean ligand RMSD is -152.76 and 46.73 Å, respectively. The docking score and ligand RMSD, as well as the important residues on SSL10 of each docking models were listed in Table R1.

The model at the top of the list was chosen for further analysis. The docking score of this model was -222.54 and ligand RMSD was 38.73 Å. The binding free energy of complex in this docking model was -13.66 kcal/mol calculated by the Hawkdock webserver. Eight residues of SSL10 (H64, K66, N85, S88, Q91, K206, K208 and Y209) predicted to be involved in the interaction with TNFR1.

Table R1. Analysis of ten docking models obtained from the HDock webserver.

Models	Important residues on interface of SSL10	Docking score	Ligand RMSD (Å)
1	H64, K66, N85, S88, Q91, K206, K208, Y209	-222.54	38.73
2	H64, K66, N85, K87, S88, K106, Y114, K208	-157.68	41.23
3	H64, K66, S88, Q91, K106, Y114, K204, K208	-152.97	28.38
4	H64, K66, N85, S88, Q91, K206, K208, Y209	-151.51	64.12
5	K66, N85, S88, Q91, R93, K188, K208, Y209	-151.18	47.82
6	H64, K66, N85, S88, Q91, K206, K208, Y209	-150.04	73.49
7	H64, K66, N85, S88, Q91, K206, K208, Y209	-147.69	41.91
8	H64, K66, N85, S88, Q91, K206, K208, Y209	-135.99	48.82
9	H64, K66, N85, S88, Q91, K206, K208, Y209	-132.38	47.68
10	K66, N67, N68, K87, R93, K106, F207, K208	-125.58	35.12

As suggested, we also performed docking analysis to obtain SSL10-TNFR1 interaction model from the Hawkdock webserver and the model with the highest scored was chosen for further analysis. The docking score and binding free energy of this model is -5383.28 and -14.19 kcal/mol, respectively. Comparison with the results obtained from HDock webserver showed that all the eight residues of SSL10 (*i.e.*, H64, K66, N85, S88, Q91, K206, K208 and Y209) predicted to be involved in the interaction with TNFR1 were identified in the docking model generated by the Hawkdock webserver. The binding interface of SSL10 and TNFR1 are positively and negatively charged, respectively (Fig. 6b and Fig. 6f). Because SSL10 and TNFR1 fit better in the model generated by the Hawkdock webserver, we replaced Fig. 6a and 6b by this model in the revised manuscript. Residue mutations or domain deletion of the interface on both SSL10 and TNFR1 resulted in a decreased binding ability (Fig. 6d, 6e and 6g-h), indicating that these residues or domain contributed to the interaction of SSL10 with TNFR1.

Figure 6

Fig. 6 Identification of potential binding surface between SSL10 and TNFR1. (a) The structures of SSL10 and TNFR1^{ECD} (PDB code: 1EXT) were docked using the Hawkdock program to generate structural models of the SSL10/TNFR1^{ECD} complex. Enlarged view shows the residues of SSL10 possibly involved in binding to TNFR1^{ECD}. SSL10 and TNFR1^{ECD} are colored cyan and light brown, respectively. Amino acid residues important for binding are shown as sticks. (b) The potential TNFR1 binding sites on SSL10. The electrostatic surface potentials of SSL10 are shown. Positively and negatively charged surface regions are colored blue and red,

respectively. (c) LDH released from HEK293T cells treated with 2 μ M SSL10 or either of the two mutants. The eight residues potentially involved in SSL10-TNFR1 interface, *i.e.*, H64, K66, N85, S88, Q91, K206, K208, and Y209 in SSL10, were replaced with either alanine residues (mutant A) or with the corresponding residues from SSL7 (mutant B). (d) Immunoblotting of wild type (WT) or mutant SSL10 after pull-down with MBP-TNFR1^{ECD}. The relative band intensities of WT or mutant SSL10 proteins pulled down by MBP-TNFR1^{ECD} are quantitated by densitometry after normalization to their input, and then expressed as the fold of WT SSL10. (e) ITC assays of mutant A and mutant B SSL10 proteins binding to MBP-TNFR1^{ECD}. (f) The potential SSL10 binding surface on TNFR1. The electrostatic surface potentials of TNFR1 are shown. Positively and negatively charged surface regions are colored blue and red, respectively. (g) Immunoblotting of SSL10 after pull-down with MBP-TNFR1^{ECD} or its four deletion mutants (*i.e.*, Δ CRD1-4). The relative band intensities of SSL10 proteins pulled down by WT or mutant MBP-TNFR1^{ECD} are quantitated by densitometry after normalization to their input, and then expressed as the fold of WT SSL10 pulled down by WT MBP-TNFR1^{ECD}. (h) ITC assays of SSL10 binding to the TNFR1^{ECD} mutant Δ CRD1 or Δ CRD2. (i) LDH released from HUVEC cells treated with 2 μ M SSL10 alone or combined with 10 μ M MBP-TNFR1^{ECD}, Δ CRD1 or Δ CRD2, as indicated. All data represent means \pm SD calculated from at least three independent experiments. *, $p < 0.05$; **, $p < 0.01$, ***, $p < 0.001$ compared to the Ctrl cells (buffer-treated cells) or as indicated. #, $p < 0.05$; ##, $p < 0.01$ compared to the SSL10 treated group.

6) Secondly, the analysis showing that the mutated forms (A or B) of SSL-10 do not interact with TNFR1 is very weak. Analysis in Fig 6D indicates mutation does not abolish necroptotic activity of SSL-10. Analysis in Fig 6E and 6F shows reduced pulldown of SSL-10 mutants compared to SSL-10 wildtype by TNFR1. Neither dataset indicate that the mutated SSL-10 residues are critical for TNFR1 interaction, otherwise signals would be 0. Thus the conclusion "Therefore, the docking and mutagenesis analyses demonstrate that 259 residues H64, K66, N85, S88, Q91, K206, K208, and Y209 are critical for SSL10 binding with TNFR1^{ECD} 260 and subsequent initiation of the necroptosis signal cascade." is not supported. Complete *in silico* analysis and further biochemical analysis is required to support this conclusion.

Response: We are sorry about the inaccurate data presentation. To further analyze the interface between SSL10 and TNFR1, we also performed docking analysis to obtain SSL10/TNFR1 interaction model using the Hawkdock webserver. Comparison with the results obtained from the HDOCK webserver showed that all the eight residues of SSL10 predicted to be involved in the interaction with TNFR1 by the HDOCK webserver were also identified in the docking model generated by Hawkdock webserver (Fig. 6a, see above in comment 5). We also performed immunoblotting and ITC assays to verify whether these eight residues are involved in the binding of SSL10 to TNFR1^{ECD}. Immunoblotting and ITC results showed that both mutant A and mutant B displayed much weaker binding ability to TNFR1^{ECD}, with the K_D values of mutant A and mutant B being 31.2 and 45 μ M, respectively (compared to that of wild type SSL10 at 3.87 μ M) (Fig. 6d and 6e, see above in comment 5). These data demonstrated that the residues H64, K66, N85, S88, Q91, K206, K208, and Y209 are important for the binding of SSL10 to TNFR1^{ECD}.

7) *Furthermore, what are the critical residues in TNFR1 required for SSL-10 interaction? what support is there for this interaction?*

Response: Based on the docking model generated by the Hawkdock webserver, the interaction of TNFR1^{ECD} with SSL10 is mediated mainly by a negatively charged region on the second Cysteine-rich domain (CRD2), one of the four CRDs in extracellular domain of TNFR1 (Fig. 6f, see above in comment 5). To verify this model, we constructed 4 TNFR1 deletion mutants (*i.e.*, Δ CRD1, Δ CRD2, Δ CRD3, and Δ CRD4, respectively) according to its extracellular cysteine-rich domains (CRDs)¹². Both immunoblotting and ITC analysis showed significantly decreased binding of TNFR1 mutant Δ CRD2 to SSL10, with K_D value of 30.3 ± 3.55 μ M (Fig. 6g and 6h, see above in comment 5). Consistently, the inhibitory effects of Δ CRD2 mutant on SSL10-induced cytotoxicity was also significantly weaker than that of TNFR1^{ECD} (Fig. 6i, see above in comment 5). These results indicate that the second CRD region of TNFR1^{ECD} is responsible for its interaction with SSL10.

8) *SSL-10 is purified from E. coli cultures. LPS endotoxins are well known to be contaminants of protein purifications and have capacity to induce necroptosis. Given the nature of the study, it is critical that LPS endotoxins are removed from SSL-10 and other protein purifications and that protein controls are used throughout the cell-based assays. What strategies have been used to eliminate LPS from protein purifications? how has this been validated? Protein controls (such as SSL7) should be used throughout the cell based assays in Fig 1 to 5.*

Response: Thank you for the suggestion. To exclude the potential effects of LPS on the cytotoxicity of SSL10, several experiments have been performed. First, we have confirmed that the all the proteins used in this study, which were purified by molecular sieve chromatography, have very little amount of LPS endotoxin (with a concentration of 0.016 ng/mL) as determined by the ToxinSensor™ Chromogenic LAL Endotoxin Assay Kit (Genscript). To further exclude a potential role of LPS in recombinant SSL10-induced necroptosis, we treated the HEK293T cells with LPS (1 ng/mL) or SSL10 in the presence of LPS blocker polymyxin B (10 μM). LPS treatment did not show any cytotoxicity, and LPS blocker did not show any impacts on SSL10-induced cytotoxicity (Fig. R4), indicating that SSL10-induced necroptosis is unlikely to be caused by LPS contamination. Furthermore, we have used SSL7, which was expressed and purified following exactly the same protocol as SSL10, as a negative protein control throughout the cell-based assays to exclude the effects of LPS (or other contamination from *E. coli*) on the necroptosis induced by recombinant SSL10 in the revised manuscript. All these data indicate that the necroptosis is specifically induced by SSL10, but not LPS or other contaminants.

Fig. R4. Effects of LPS and LPS blocker on the cell variability. HEK293T cells were treated with LPS (1 ng/mL) or SSL10 (2 μ M) in the absence or presence of LPS blocker polymyxin B (10 μ M). **(a)** The release of LDH was detected. **(b)** The cell viability was measured by flow cytometry after Annexin V/PI staining, and the results of quantification are shown as a bar graph. Cells treated with 2 μ M SSL7 were used as a negative protein control throughout the experiments. All data represent means \pm SD calculated from at least three independent experiments. **, $p < 0.01$; ***, $p < 0.001$ compared to the Ctrl cells (buffer- treated cells).

9) *what is the predicted interface of SSL-10 for other ligands? has this been characterised in previous studies? what does in silico analysis show?*

Response: We used Hawkdock webserver to generate the docking models of SSL10 with IgG, CXCR4 and prothrombin. The docking models with highest score were shown in Fig. R5. SSL10 was predicted to bind IgG through N-terminal OB fold, and interact with prothrombin via both the N-terminal OB fold and C-terminal β -grasp domain according to the docking models, respectively. This is consistent with the binding regions of SSL10 to different ligands mapped by biochemical analysis^{8,9}. In addition, both the N-terminal OB fold and C-terminal β -grasp domain of SSL10 were predicted to binds CXRC4, as suggested by the docking model (Fig. R5).

Fig. R5. The docking models of SSL10 in complex with CXCR4 (PDB: 3ODU), IgG1 Fc (PDB: 1H3X) and prothrombin (PDB: 5EDM). **a-b**, Docking model of SSL10/CXCR4 complex (**a**) and the residues predicted for interaction with CXCR4 on SSL10 interface (**b**). **c-d**, Docking model of SSL10/IgG1 Fc complex (**c**) and the residues predicted for interaction with IgG1 Fc on SSL10 interface (**d**). **e-f**, Docking model of SSL10/prothrombin (**e**) and the residues predicted for interaction with prothrombin on SSL10 interface (**f**). SSL10, CXCR4, IgG1 Fc, and prothrombin are colored cyan, yellow, magenta, and green, respectively. Residues which are predicted to be important for binding on SSL10 are shown as sticks and labeled beside. The predicted interface on SSL10 was circled in red.

10) line 62 and 77 - given that much of the manuscript identifies a new *S. aureus* molecule that induces necroptosis, more background information is required that describes which *S. aureus* molecules are currently known to induce necroptosis and their mechanisms of action.

Response: Thank you for the suggestions. Several *S. aureus* components, including

PVL, α -toxin and PSMs, have been reported to induce necroptosis in an MLKL-dependent manner¹. In addition, *S. aureus* small colony variants can induce necroptosis by activating host cell glycolysis¹³. The additional background information has been added accordingly (Line 84-91 in the revised manuscript).

11) line 86 - change to staphylococcal superantigens?

Response: We have corrected it as suggested.

12) lines 87-88 is inaccurate. The 14 *ssl* genes are encoded at two different loci - not 1 pathogenicity island. Do all strains have genes encoding all SSL molecules? Is the gene encoding SSL-10 present in all strains?

Response: Sorry for the mistake. The 14 *ssl* genes are encoded at two different loci, with *ssl1-ssl11* in the staphylococcal pathogenicity island 2 (SaPI2), and *ssl12-ssl14* in the immune evasion cluster 2 (IEC2)¹⁴. The *ssl* gene cluster, but not all *ssl* genes, has been found in each human and animal isolate of *S. aureus*. SSL10 shows strong sequence conservation in all clinical isolates of *S. aureus* and has been found in every human and animal isolate of *S. aureus* examined so far^{8,15,16}. These points have been added in Line 98-102 in the revised manuscript.

13) line 97 - what makes SSL10 "novel" compared to other SSLs? I thought it was one of the best characterised SSLs, and therefore probably the least novel?

Response: We're sorry that our improper wording caused a misunderstanding. The "novel" we meant here is to describe SSL10 is different from other SSLs, but does not mean it is a newly identified protein. We have change this word to "unique" in the revised manuscript.

14) lines 119-120. The design of the study is not well described. What was the justification for studying *ssl3*, 7, 8, 10 and 11 only?

Response: We have tried to express and purify all of the 14 SSL proteins. However, the five SSL proteins, including SSL3, 7, 8, 10, and 11, were the first proteins to

successfully expressed and purified. Therefore, we only tested the effects of these five SSL proteins.

15) Fig 1A and B - there is no protein control in these assays. Surely an alternative SSL could be used as a better control than buffer?

Response: Thanks for your suggestion. SSL7, which was expressed and purified following exactly the same protocol as SSL10, was used as a negative protein control throughout the cell based assays in the revised manuscript.

16) lines 130-132. There is no data presented that demonstrates SSL10 is present in the *S. aureus* culture supernatants - or the quantity. How do you know SSL10 is expressed by 8235-4 and released into the supernatant?

Response: Thank you for the suggestion. Since commercial antibodies for SSL10 are unavailable, we couldn't directly demonstrate that there was SSL10 in the supernatants. To address this question, we constructed an 8325-EGFP strain in which the gene *egfp* was fused to the C-terminus of *ssl10* (Fig. R2a). SSL10-EGFP can be clearly detected by immunoblotting using anti-GFP antibody, indicating the presence of SSL10 in the supernatant (Fig. R2b).

Fig. R2. (a) Genomic schematic diagram of NCTC 8325 and NCTC 8325-EGFP strain. (b)

SSL10-EGFP detection in the supernatant of NCTC 8325 and 8325-EGFP strains with anti-GFP antibody by immunoblotting.

17) Fig2A and 2B - there looks to be a marginal if at all a difference between Nec1 and GSK treatments. the results here are overstated

Response: We do agree that there was no significant difference between Nec-1 and GSK872', which may be caused by the off-target effects of Nec-1. Nec-1 is the first discovered necrostatin and acts as an allosteric inhibitor of the kinase domain of RIPK1. However, Nec-1 has been shown to have off-target effects by inhibiting indoleamine 2,3-dioxygenase (IDO), a potent immunomodulatory enzyme¹⁷. To exclude the off-target effects, we also used a more specific RIPK1 inhibitor Nec-1 stable (Nec-1s), which showed apparent difference compared to GSK treatments. To avoid any confusion, we have deleted the data of Nec-1 treatments in Fig. 2a in the revised manuscript. As shown in Fig. 2a, the inhibitory effect of GSK'872 is significantly potent than that of Nec-1s.

Fig. 2a. HEK293T and HUVEC cells pretreated with 50 μ M Nec-1s or 50 μ M GSK'872 for 30 min, and then stimulated with 2 μ M SSL10 for 48 h at 37 $^{\circ}$ C and 5% CO_2 . The release of LDH was detected. Cells treated with 2 μ M SSL7 were used as a negative protein control throughout the experiments. All data represent means \pm SD calculated from at least three independent experiments. *, $p < 0.05$; **, $p < 0.01$; ***, $p < 0.001$ compared to the Ctrl cells (buffer- treated cells). #, $p < 0.05$; ##, $p < 0.01$; ###, $p < 0.001$ compared to the SSL10 treated group.

18) lines 183-184. which other receptors induce necroptosis? what was the justification for studying TNFR1 as opposed to any other receptor?

Response: Necroptosis can be triggered via several different membrane receptors including TNFR1, toll-like receptors (TLR3 and TLR4), and interferon receptors¹⁸. To test which membrane receptor is in response to SSL10, the inhibitors of TLR4 (TAK-242) and IFNAR1 (IFN alpha-IFNAR-IN-1) were used to detect their roles in the SSL10-induced necroptosis. We found that pretreatment of cells with TAK-242 or IFN alpha-IFNAR-IN-1 has no effect on the cytotoxicity induced by of SSL10 (Fig. R6). However, the cytotoxicity of SSL10 on *tnfrsf1a* knockout cell line significantly decreased, to a level that has no significant difference from that of cells treated with buffer or SSL7 (Fig. 4b-d). Thus we speculate that SSL10 may trigger necroptosis mainly through TNFR1.

Fig. R6 Necroptosis induced by SSL10 is not dependent on TLR4 or IFNAR1. HEK293T cells were pretreated with 10 μ M TAK-242 or 1 μ M IFN alpha-IFNAR-IN-1 (IFNAR-IN) for 1 h followed by SSL10 treatment for 48 h, and the LDH release was detected. All data are presented as the mean \pm SD from at least three independent experiments. ***, $p < 0.001$ compared to the Ctrl cells (buffer-treated cells), n.s., not significant, by multiple t-test.

Fig. 4 SSL10 induces necroptosis by direct interacting with the extracellular domain of TNFR1 (TNFR1^{ECD}). (b) WT or TNFR1 KO cells were treated with 2 μ M SSL10 for 48 h at 37 $^{\circ}$ C and 5% CO₂, and the LDH release was detected. (c) WT or TNFR1 KO HEK293T cells were challenged with SSL10 for 48 h and the cell viability was measured by flow cytometry after Annexin V/PI staining. The dot plot (left) is representative of three independent experiments, and the results of quantification (right) are shown as a bar graph. (d) Depolarization of the mitochondrial membrane of WT or TNFR1 KO HEK293T cells treated by SSL10 was measured by flow cytometry after JC-1 staining. Cells treated with 2 μ M SSL7 were used as a negative protein control throughout the experiments. All data represent means \pm SD calculated from at least three independent experiments. **, $p < 0.01$; ***, $p < 0.001$ compared to the Ctrl cells (buffer-treated cells). #, $p < 0.05$; ##, $p < 0.01$; ###, $p < 0.001$ compared to the SSL10 treated group.

19) why does TNFR1 knockdown only partly suppress SSL10 induced necroptosis?

Fig 4C

Response: We do agree that there is a little residual cytotoxicity induced by SSL10 treatment in TNFR1 knockout cells, but there's no significant difference from that of buffer or SSL7 treated cells (Fig. 4b-d, see above in comment 18). In addition, anti-TNFR1 mAb and purified MBP-tagged TNFR1 extracellular domain (TNFR1^{ECD}) impaired SSL10-induced necroptosis of HUVEC in a dose-dependent manner (Fig. S5a and S5b, see above in comment 1). These data indicate that SSL10 mainly triggered necroptosis via interacting with TNFR1. Of course we cannot exclude the possibility that SSL10 can also induce cell death via other unknown mechanisms, since there are still some SSL10 proteins bound to the cell surface of TNFR1 knockout cells (Fig. 4a), although such effects are not facilitated by other known receptors, including TLR4 or IFNAR1 (Fig. R6, see above in comment 18).

Fig. 4a. Wild type (WT) or TNFR1 knockout (KO) HEK293T cells were treated with SSL10 at different concentrations for 30 min, and the SSL proteins bound to the cell surface were detected by flow cytometry after FITC-conjugated anti-His-tag antibody staining. All data represent means \pm SD calculated from at least three independent experiments. *, $p < 0.05$; **, $p < 0.01$; ***, $p < 0.001$ compared to the Ctrl cells (buffer-treated cells) or as indicated, ##, $p < 0.01$ compared to the SSL10 treated group.

20) data presentation in Fig 4 is a bit backwards to me. It would be best to show reduced binding of SSL10-GFP to knockdown cells (F), then reduced LDH release (B), increased cell viability (D, E)

Response: Thank you for the suggestion. Data in Fig. 4 have been re-organized as suggested.

21) *Fig 6A. The individual residues to be highlighted cannot be seen. They should be presented in an alternative colour to the remainder of the chain.*

Response: Sorry about the negligence. We have changed the color of residues highlighted on SSL10 in Fig. 6a in the revised manuscript.

22) *Fig 6B. Residues of interest cannot be visualised.*

Response: We have changed the color of residues highlighted on SSL10 in Fig. 6b in the revised manuscript.

23) *Fig 6C. SSL-7 and SSL-10 are both variable. The sequences in the alignment are from which genome? Do any SSL10 variants have mutations are the residues identified? Do any SSL7 variants have the residues identified for TNFR1 interaction?*

Response: The sequences in the alignment are from *S. aureus* Mu50. We aligned SSL10 and SSL7 sequences from 9 *S. aureus* strains, including Mu50, NCTC8325, N315, NCTC10654, Newman, USA300, MRSA252, NCTC7878, and MN8. Among these strains, five identified residues (*i.e.*, H64, K66, Q91, K206, and K208) are conserved among all the strains, while the other three residues (*i.e.*, N85, S88, and Y209) vary among different strains (Fig. R7a). Sequence alignment of SSL7 from various *S. aureus* strains showed that none of the residues proposed to be involved in the interaction between SSL10 and TNFR1 was present in SSL7 (Fig. R7b).

Fig. R7 Sequence alignment of SSL10 and SSL7 from various *S. aureus* strains. Sequence alignment of SSL10 (a) and SSL7 (b) from *S. aureus* strains Mu50, NCTC8325, N315, NCTC10654, Newman, USA300, MRSA252, NCTC7878 and MN8 was conducted with MultAlin and ESPrpt. Amino acid residues of SSL10 from Mu50 possibly involved in the interaction with TNFR1^{ECD} are indicated by blue arrows. The residues different from that identified are labeled by blue asterisk. Sequence identities of SSL10 (or SSL7) from Mu50 to SSL10 (or SSL7) from other strains calculated by Clustal Omega are listed below each figure.

References:

- 1 Kitur, K. *et al.* Toxin-induced necroptosis is a major mechanism of *Staphylococcus aureus* lung damage. *PLoS Pathog* **11**, e1004820, doi:10.1371/journal.ppat.1004820 (2015).
- 2 Zhou, Y. *et al.* Inhibiting PSM α -induced neutrophil necroptosis protects mice with MRSA pneumonia by blocking the agr system. *Cell Death Dis* **9**, 362, doi:10.1038/s41419-018-0398-z (2018).
- 3 Wang, H. *et al.* Mixed lineage kinase domain-like protein MLKL causes necrotic membrane disruption upon phosphorylation by RIP3. *Mol Cell* **54**, 133-146, doi:10.1016/j.molcel.2014.03.003 (2014).
- 4 Oliver Metzger, M. *et al.* Inhibition of caspases primes colon cancer cells for 5-fluorouracil-induced TNF- α -dependent necroptosis driven by RIP1 kinase and NF- κ B. *Oncogene* **35**, 3399-3409, doi:10.1038/onc.2015.398 (2016).
- 5 Robinson, N. *et al.* Type I interferon induces necroptosis in macrophages during infection with *Salmonella enterica* serovar Typhimurium. *Nat Immunol* **13**, 954-962, doi:10.1038/ni.2397 (2012).
- 6 den Reijer, P. M. *et al.* Characterization of the humoral immune response during *Staphylococcus aureus* bacteremia and global gene expression by *Staphylococcus aureus* in human blood. *PLoS One* **8**, e53391, doi:10.1371/journal.pone.0053391 (2013).
- 7 Kastritis, P. L. *et al.* A structure-based benchmark for protein-protein binding affinity. *Protein Sci* **20**, 482-491, doi:10.1002/pro.580 (2011).
- 8 Patel, D., Wines, B. D., Langley, R. J. & Fraser, J. D. Specificity of staphylococcal superantigen-like protein 10 toward the human IgG1 Fc domain. *J Immunol* **184**, 6283-6292, doi:10.4049/jimmunol.0903311 (2010).
- 9 Itoh, S., Takii, T., Onozaki, K., Tsuji, T. & Hida, S. Identification of the blood coagulation factor interacting sequences in staphylococcal superantigen-like protein 10. *Biochem Biophys Res Commun* **485**, 201-208, doi:10.1016/j.bbrc.2017.02.053 (2017).
- 10 Langley, R. *et al.* The staphylococcal superantigen-like protein 7 binds IgA and complement C5 and inhibits IgA-Fc α RI binding and serum killing of bacteria. *J Immunol* **174**, 2926-2933, doi:10.4049/jimmunol.174.5.2926 (2005).
- 11 Itoh, S. *et al.* Staphylococcal superantigen-like protein 10 (SSL10) inhibits blood coagulation by binding to prothrombin and factor Xa via their gamma-carboxyglutamic acid (Gla) domain. *J Biol Chem* **288**, 21569-21580, doi:10.1074/jbc.M113.451419 (2013).
- 12 Kimberley, F. C., Lobito, A. A., Siegel, R. M. & Sreaton, G. R. Falling into TRAPS--receptor misfolding in the TNF receptor 1-associated periodic fever syndrome. *Arthritis Res Ther* **9**, 217, doi:10.1186/ar2197 (2007).
- 13 Wong Fok Lung, T. *et al.* *Staphylococcus aureus* small colony variants impair host immunity by activating host cell glycolysis and inducing necroptosis. *Nat Microbiol* **5**, 141-153, doi:10.1038/s41564-019-0597-0 (2020).
- 14 Kuroda, M. *et al.* Whole genome sequencing of methicillin-resistant *Staphylococcus aureus*. *Lancet* **357**, 1225-1240, doi:10.1016/s0140-6736(00)04403-2 (2001).
- 15 Smyth, D. S., Meaney, W. J., Hartigan, P. J. & Smyth, C. J. Occurrence of ssl genes in isolates of *Staphylococcus aureus* from animal infection. *J Med Microbiol* **56**, 418-425,

doi:10.1099/jmm.0.46878-0 (2007).

- 16 Fitzgerald, J. R. *et al.* Genome diversification in *Staphylococcus aureus*: Molecular evolution of a highly variable chromosomal region encoding the Staphylococcal exotoxin-like family of proteins. *Infect Immun* **71**, 2827-2838, doi:10.1128/IAI.71.5.2827-2838.2003 (2003).
- 17 Mikus, P. *et al.* Determination of Novel Highly Effective Necrostatin Nec-1s in Rat Plasma by High Performance Liquid Chromatography Hyphenated with Quadrupole-Time-of-Flight Mass Spectrometry. *Molecules* **23**, doi:10.3390/molecules23081946 (2018).
- 18 Pasparakis, M. & Vandenabeele, P. Necroptosis and its role in inflammation. *Nature* **517**, 311-320, doi:10.1038/nature14191 (2015).

Reviewers' comments:

Reviewer #1 (Remarks to the Author):

The manuscript is much improved and my concerns have been addressed. It contains numerous grammatical errors and "wide" is used instead of "wild" in lines 87 and 276. In Line 513 "DNA fragment of human TNFR1 extracellular domain (amino acid residues 22-211) was amplified by PCR". The source of the human DNA needs to be specified as well as how the identity of the cloned gene was verified (was it sequenced?).

Reviewer #3 (Remarks to the Author):

The manuscript is an improvement on the initial submission. However, there still remains an issue with molecular docking. In particular, the simulated docking of the *S. aureus* SSL10 molecule to TNFR1 is not scientifically sound yet. The quality of the models is questionable. Please see below. In addition, identifying the interface residues is not scientifically sound – this needs to be assessed across many iterations so that the contribution of each residue to the binding interface can be calculated. This can be assessed in the MM/GBSA component of the HawkDock server. In addition, a number of comments I previously raised have been 1) addressed well in the response but not in the manuscript, 2) or incorrectly addressed in the text.

Major Comments

1. Modelling the SSL10-TNFR interaction in Hawkdock. Results & Discussion. I think this analysis requires further attention. This is related to comment 5 in my original review. In computational biology, many different models must be generated in order to develop confidence in a model generated. The free binding energy should be provided for docked models that can be used to assess the likelihood that a model does show the docking of a molecule to its receptor. This is now provided in the rebuttal as 46.73 Å for the model analysed and presented in Hdock. However, the authors have not considered what this score means. It is generally accepted that a RMSD ≤ 2.0 Å is indicative of a good solution of the docking, a score of between 2.0 and 3.0 Å as acceptable, and a score of > 3.0 Å as a bad solution. A score of 46.73 Å would suggest that the model generated is of poor quality, unreliable and wrong. Furthermore, the data in Table R1 would indicate that all the models generated in Hdock have a high RMSD score and are unreliable. The authors have now modelled the docking using HawkDock with an score of -14.19 kcal/mol. This is generally lower than a cut-off value (of -15) and is therefore borderline accurate/inaccurate. With this said, readers should be presented with information to interpret the accuracy of the model and the authors should discuss the accuracy of their models at an appropriate stage of the manuscript at a minimum.
2. In addition, to identify binding interfaces in a predicted model- it is not appropriate to analyse one model. Instead, multiple models must be generated and the mean binding free energy contributions of each residue across the models should be calculated. As rough guide, residues with binding free energy contributions lower than -2.0 kcal/mol are usually identified as key residues. This work requires more thorough analyses to really ascertain the binding interface and to be publishable.
3. Responses to original comments are not addressed satisfactorily in the manuscript – some of which are outlined above and below. Comments 5, 12, 13, 14, 18. Some of these responses are wrong. Some of the responses (comment 14, 18) are explained in response but not addressed in the revised MS.

Minor Comments

1. Lines 51-53. The language in sentence "identified a surface region formed by residues in the N- and C-termini that can potentially serve as the binding site for TNFR1 via its second extracellular cysteine-rich domain (CRD)" is very elusive and non-scientific. In particular, the use of "potentially" is not appropriate. I suggest this sentence is revised.
2. Line 54. "Profoundly" is inappropriate language. Please modify.

3. Line 56-67. "by an SSL protein" please be specific..."by the SSL10 protein".
4. Line 69. "S. aureus can manipulate host immune response through the expression of a myriad" should be modified "S. aureus can manipulate the host immune response through the expression of a myriad"
5. Line 72. Given the importance of necroptosis to this study, I think virulence factors known to induce necroptosis should be stated here.
6. Line 99-101. "The 14 ssl genes are encoded at two different loci, with ssl1-ssl11 in the staphylococcal pathogenicity island 2 (SaPI2), and ssl12-ssl14 in the immune evasion cluster." This is incorrect. The genes are not encoded on SaPI2. They are encoded on a genomic island NOT a pathogenicity island. Please correct.
7. Lines 102-105. "Among all the SSL proteins, SSL10 is a unique member, which has been found in every human and animal isolate of S. aureus21-24, is involved in several pathological process." This is incorrect. SSL10 gene is not found in some genomes. Please see PMID: 23792184. Please clarify why "SSL10 is a unique member".
8. Line 124. Please justify selection of analysis of "SSL3, SSL7, SSL8, SSL10 and SSL11" but not other SSLs
9. Lines 143-145. Please perform statistical analysis of data points on Fig 1F and include results on the figure or include text that states complemented mutant was or was not significant different from the mutant strain.
10. Fig 4. Text in Fig 4F is too small. The Figure needs to be bigger.
11. Line 235 "Furthermore, differ from other SSL proteins, several," does not make sense. Please modify.
12. Line 237-238. "The distinct structural feature might endow SSL10 with specificity to interact with TNFR1." This statement is not justified. Please justify or remove.
13. Line 264-269. "The sequence identity between SSL10 and its homologs was relatively low, approximately 30-40% (Fig. S8). More significantly, the eight residues identified above in SSL10 differed from the corresponding residues in other SSL proteins on charge, while these corresponding residues were more similar among other SSL proteins, indicating the specificity of SSL10 to interact with TNFR1ECD, which has more negatively charged surface." These statements are not justified. It cannot be said that the difference in these 8 residues determines the TNFR1ECD specificity without mechanistic assessment when there is such extensive variation between the molecules. Likewise, there is no evidence provided that charge of molecules determines the TNFR1ECD specificity – this would require mutation of residues to opposite charges. Please revise, justify or remove.
14. Lines 279-282. "Therefore, the docking and mutagenesis analyses demonstrate that residues H64, K66, N85, S88, Q91, K206, K208, and Y209 are involved in SSL10 binding with TNFR1ECD and subsequent initiation of the necroptosis signal cascade." This is not justified in the context of docking – please see above comments.
15. Line 307. Correct to HEK293.

Point-to-point Response Letter

Please find our responses to the specific comments raised by the reviewers below. We have copied each comment in *Italic*, which is followed by our point-by-point response in blue. Proper changes have been incorporated into the revised manuscript with all changes being highlighted.

Reviewer #1 (Remarks to the Author):

The manuscript is much improved and my concerns have been addressed.

1. It contains numerous grammatical errors and "wide" is used instead of "wild" in lines 87 and 276.

Response: Sorry for the mistakes. We have corrected the typos in the manuscript.

2. In Line 513 "DNA fragment of human TNFR1 extracellular domain (amino acid residues 22-211) was amplified by PCR". The source of the human DNA needs to be specified as well as how the identity of the cloned gene was verified (was it sequenced?).

Response: Thanks for your suggestion. We changed the sentence to “The DNA fragment of human TNFR1 extracellular domain (amino acid residues 22-211) was amplified by PCR using cDNA library of human spinal cord as template and cloned into the pET-28a (+) vector with an N-terminal MBP-tag, which was verified by Sanger sequencing.” (line 491-497).

Reviewer #3 (Remarks to the Author):

*The manuscript is an improvement on the initial submission. However, there still remains an issue with molecular docking. In particular, the simulated docking of the *S aureus* SSL10 molecule to TNFR1 is not scientifically sound yet. The quality of the models is questionable. Please see below. In addition, identifying the interface residues is not scientifically sound – this needs to be assessed across many iterations so that the contribution of each residue to the binding interface can be calculated. This can be assessed in the MM/GBSA component of the HawkDock server. In addition, a number of comments I previously raised have been 1) addressed well in the response but not in the manuscript, 2) or incorrectly addressed in the text.*

Major Comments

1. Modelling the SSL10-TNFR interaction in Hawkdock. Results & Discussion. I think this analysis requires further attention. This is related to comment 5 in my original review. In computational biology, many different models must be generated in order to develop confidence in a model generated. The free binding energy should be provided for docked models that can be used to assess the likeliness that a model does show the docking of a molecule to its receptor. This is now provided in the rebuttal as 46.73 Å for the model analysed and presented in Hdock. However, the authors have not considered what this score means. It is generally accepted that a RMSD ≤ 2.0 Å is indicative of a good solution of the docking, a score of between 2.0 and 3.0 Å as acceptable, and a score of > 3.0 Å as a bad solution. A score of 46.73 Å would suggest that the model generated is of poor quality, unreliable and wrong. Furthermore, the

data in Table R1 would indicate that all the models generated in Hdock have a high RMSD score and are unreliable. The authors have now modelled the docking using HawkDock with an score of -14.19 kcal/mol. This is generally lower than a cut-off value (of -15) and is therefore borderline accurate/inaccurate. With this said, readers should be presented with information to interpret the accuracy of the model and the authors should discuss the accuracy of their models at an appropriate stage of the manuscript at a minimum.

Response: Thanks for your suggestion. Because the docking model predicted by Hdock had a relatively high RMSD score and might be unreliable, we performed docking analysis of SSL10 and TNFR1^{ECD} (PDB code: 1EXT) using the HawkDock webserver (<http://cadd.zju.edu.cn/hawkdock/>). The binding interfaces of the top 10 predicted models were further analyzed by Molecular Mechanics/Generalized Born Surface Area (MM/GBSA), and the docking score as well as the binding free energy of each model were listed in Table S3.

Table S3. The docking score and binding free energy of the top 10 SSL10/TNFR1^{ECD} models generated by HawkDock.

Model	Docking score	Binding free energy (kcal/mol)
1	-5383.28	-13.16
2	-4934.31	-22.02
3	-4780.94	-7.51
4	-4340.73	-22.34
5	-4325.42	-37.63
6	-4241.46	-14.11
7	-4039.77	-11.18
8	-4001.47	-22.29
9	-3967.16	-30.28
10	-3959.68	-28.98

We next analyzed the reliability of these models based on the previously obtained biochemical data (Figs. 5 and 6), which showed that both the N- and C-terminal domains of SSL10 contribute to the interaction with TNFR1^{ECD}, and the recognition of SSL10 is largely mediated by CRD2 of TNFR1^{ECD}. Accordingly, six models were excluded because the predicted binding interface was not consistent with the biochemical data, and only model 1, 2, 4, and 8 were included for further analysis (Fig. S9a). In each model, TNFR1^{ECD} binds to a positively charged region on SSL10, suggesting the recognition of TNFR1 by SSL10 is dominantly mediated by electrostatic interactions (Figs. 6d-e, S9b).

Fig. 6 Identification of a potential binding interface between SSL10 and TNFR1. (d) Model 1 (M1) of SSL10/TNFR1^{ECD} complex was generated by HawkDock program with the crystal structures of SSL10 and TNFR1^{ECD} (PDB code: 1EXT). The enlarged view shows the residues of SSL10 predicted to interact with TNFR1^{ECD}. SSL10 and TNFR1^{ECD} are colored cyan and light brown, respectively. (e) The electrostatic surface view of the predicted binding sites of SSL10 for TNFR1^{ECD} is shown. The interface residues on SSL10 for TNFR1 binding are shown as sticks and labeled in black. The positive and negative charge are colored blue and red, respectively.

Supplemental Figure 9

Fig. S9 The top 10 models of SSL10/TNFR1^{ECD} complex generated by the HawkDock webserver. **(a)** Cartoon presentation of SSL10/TNFR1^{ECD} complex models. The N-terminal domain of SSL10, the C-terminal domain of SSL10, the CRD2 region of TNFR1^{ECD}, and the other regions of TNFR1^{ECD} were colored in green, orange, magenta, and light brown, respectively. **(b)** The electrostatic surface potentials of SSL10 molecules from Model 2, 4 and 8. The interface residues on SSL10 responsible for TNFR1 binding are shown as sticks and labeled in black. The positive and negative charge are colored blue and red, respectively.

To further assess the reliability of these our models, the residues on SSL10 predicted to form the binding interface with TNFR1 were replaced by either amino acids with

opposite charge or their counterparts in SSL3, SSL7, SSL8 or SSL11 (Table S4). All mutant SSL10 proteins (*i.e.*, M1, M2, M4, and M8) were expressed in *E. coli* and purified (Fig. S10), and the binding affinity of the well-folded proteins (M1 and M2) to TNFR1 and the cytotoxicity of M1 to HEK293T cells were determined (Fig. 6f-h). The binding of M1 but not M2 to TNFR1 was abolished and cytotoxicity of M1 was significantly hampered, indicating that a binding surface on SSL10 predicated by model 1 were responsible for TNFR1 interaction and necroptosis induction. We have described and discussed these data in line 254-278 in the revised manuscript.

Table S4. Mutated residues in SSL10 mutants generated according to the predicted model 1, 2, 4 and 8 of SSL10/TNFR1^{ECD} complex.

SSL10 Mutants	Mutated residues
M1	H64D, N67R, N68K, R70E, K106D, K206D, F207E, K208E, Y209R
M2	K52E, T53S, M54F, E55K, R74E, I76Q, K166E, K169E
M4	R74E, K147E, Y149D, Y152L, K154D, K166E, K169E, H170D, E173K
M8	N67R, N68K, Q79K, K87E, K106E, Y114S, K204E, K208E

Supplemental Figure 10

Fig. S10 Purification of SSL10 mutants. (a) Multiple sequence alignment of SSL10 with SSL3ΔN, SSL7, SSL8, and SSL11 generated by MultAlin and ESPrpt. The secondary structure elements of SSL10 are shown above the sequences. Sequence identities of SSL3ΔN, SSL7, SSL8, and SSL11 to SSL10 are calculated by Clustal Omega and shown at the bottom. SSL10 residues desired for mutation for Model 1, 2, 4, and 8 are indicated by blue arrows, purple circles, green triangles, and orange stars, respectively. (b) Gel filtration of wild type and mutant SSL10 (M1, M2, M4, and M8) by using the Superdex75 (10/300) column (GE Healthcare). The purified proteins were verified by SDS-PAGE.

2. In addition, to identify binding interfaces in a predicted model- it is not appropriate to analyse one model. Instead, multiple models must be generated and the mean binding

free energy contributions of each residue across the models should be calculated. As rough guide, residues with binding free energy contributions lower than -2.0 kcal/mol are usually identified as key residues. This work requires more thorough analyses to really ascertain the binding interface and to be publishable.

Response: Thanks for your suggestion. As described above in the response to Comment #1, 10 models were generated by the HawkDock webserver, and Model 1 was further proved to be the most reliable one, with the binding energy being -13.16 kcal/mol. The binding energies of each residue on SSL10 predicted to interact with TNFR1 in Model 1 were also calculated (Table R1).

Table R1. The binding free energy of each residue on SSL10 predicted to interact with TNFR1 in Model 1.

Residues	Binding free energy (kcal/mol)
H64	-0.07
N67	-3.78
N68	-3.57
R70	1.4
K106	2.72
K206	-1.07
F207	-2.55
K208	-1.65
Y209	-0.05

In Model 1, five residues have the binding free energy lower than -2.0 kcal/mol: N67 (-3.78 kcal/mol), N68 (-3.57 kcal/mol), L81 (-3.33 kcal/mol), Y114 (-3.85 kcal/mol), and F207 (-2.55 kcal/mol). Among the five residues, three (N67, N68, and F207) are predicted to have hydrogen-bond interaction with TNFR1 and have been mutated in M1 mutant of SSL10. The other two residues (L81 and Y114) are predicted

to interact with TNFR1 via hydrophobic interaction and van der Waals interaction, and are not included in M1. Therefore, to explore whether these residues are critical for SSL10-TNFR1 interaction, we constructed an SSL10 M1-2 mutant with N67R, N68K, L81F, Y114S, and F207E mutation and detected its binding ability to TNFR1^{ECD}. As shown in Fig. R1, M1-2 and WT showed comparable binding ability to TNFR1^{ECD}, while M1 had little interaction with TNFR1^{ECD}. This result suggests that some residues with the binding free energy higher than -2.0 kcal/mol listed in Table R1 might be also important for SSL10-TNFR1 interaction. To avoid overstatement of the proposed model, we just described the interface but not key residues of SSL10 for TNFR1 interaction in the revised manuscript.

Fig. R1 The binding of SSL10 mutant to TNFR1^{ECD}. Immunoblotting of wild type (WT) or M1 and M1-2 mutants of SSL10 after pull-down by MBP-TNFR1^{ECD}.

3. Responses to original comments are not addressed satisfactorily in the manuscript – some of which are outlined above and below. Comments 5, 12, 13, 14, 18. Some of these responses are wrong. Some of the responses (comment 14, 18) are explained in response but not addressed in the revised MS.

Response: Thanks for the suggestions. The responses to these comments are as follows:

Comment 5 was re-addressed during this round of revision (please refer above).

Comment 12 and 13: We have revised the description for SSL proteins in line 92-96:

“The 14 *ssl* genes are encoded at two different loci, with *ssl1-ssl11* in the genomic island *vSaα*, and *ssl12-ssl14* in the immune evasion cluster 2. Among all the SSL proteins, SSL10 is a well-studied member that has been found in most human and animal isolates of *S. aureus* and involved in several pathological processes.”

Comment 14: We have added the reason for studying SSL3, 7, 8, 10, and 11 in the

Results section: “To determine the cytotoxicity of SSLs on host cells, we cloned and expressed all of the 14 SSL members in *E. coli*. SSL3, SSL7, SSL8, SSL10, and SSL11 proteins were successfully purified with high quality for cytotoxicity assays. Human umbilical vein endothelial cells (HUVEC) were treated with these purified recombinant SSL proteins for different time periods. As determined by MTS assay (Fig. S1), SSL10, but not SSL3, SSL7, SSL8 or SSL11, significantly reduced the cell activity of HUVEC after 2-day treatment.” (line 113-119).

Comment 18: We have added the reason for studying TNFR1 in the Results section and

Fig. S5: “To further determine which membrane receptor was involved in SSL10-induced necroptosis, inhibitors against TLR4 (TAK-242) and interferon receptor IFNAR1 (IFN alpha-IFNAR-IN-1) were used to pre-treat HEK293T cells before SSL10 exposure. However, SSL10-induced necroptosis was not affected by these inhibitors, indicating that SSL10-induced necroptosis is independent of TLR4 or IFNAR1 (Fig. S5).” (line 187-191).

Minor Comments

1. Lines 51-53. The language in sentence “identified a surface region formed by residues in the N- and C-termini that can potentially serve as the binding site for TNFR1 via its second extracellular cysteine-rich domain (CRD)” is very elusive and non-scientific. In particular, the use of “potentially” is not appropriate. I suggest this sentence is revised.

Response: Thanks for the suggestion. We have revised it as “We determined the crystal structure of SSL10 at 1.9 Å resolution and identified a positively charged surface of SSL10 responsible for TNFR1 binding and cytotoxic activity.” (line 45 to 47).

2. Line 54. “Profoundly” is inappropriate language. Please modify.

Response: Thanks for the suggestion. We changed “profoundly” to “significantly” in the revised manuscript (line 43).

3. Line 56-67. “by an SSL protein” please be specific... ”by the SSL10 protein”.

Response: Thank you. We revised the sentence as suggested (line 48).

4. Line 69. “*S. aureus* can manipulate host immune response through the expression of a myriad” should be modified “*S. aureus* can manipulate the host immune response through the expression of a myriad”

Response: Thank you. We revised the sentence as suggested (line 61-62).

5. Line 72. *Given the importance of necroptosis to this study, I think virulence factors known to induce necroptosis should be stated here.*

Response: Thanks for your suggestion. The information of *S. aureus* virulence factors known to induce necroptosis is summarized in line 77-81.

“Several virulence factors secreted by *S. aureus* have been shown to induce necroptosis of the host immune cells. For example, *S. aureus* toxins including Hla, PSM, LukAB, and PVL can induce RIPK1/RIPK3/MLKL-dependent necroptosis in macrophages, while PSM α is also reported to trigger neutrophil necroptosis mediated by MLKL.”

6. Line 99-101. *“The 14 ssl genes are encoded at two different loci, with ssl1-ssl11 in the staphylococcal pathogenicity island 2 (SaPI2), and ssl12-ssl14 in the immune evasion cluster.” This is incorrect. The genes are not encoded on SaPI2. They are encoded on a genomic island NOT a pathogenicity island. Please correct.*

Response: Thank you for pointing out the mistake. We have changed the sentence to “The 14 *ssl* genes are encoded at two different loci, with *ssl1-ssl11* in the genomic island vSa α , and *ssl12-ssl14* in the immune evasion cluster 2.” (line 92-93).

7. Lines 102-105. *“Among all the SSL proteins, SSL10 is a unique member, which has been found in every human and animal isolate of S. aureus²¹⁻²⁴, is involved in several pathological process.” This is incorrect. SSL10 gene is not found in some genomes. Please see PMID: 23792184. Please clarify why “SSL10 is a unique member”.*

Response: Thank you for pointing out the mistake. We have changed “a unique member”

to “a well-studied member” and revised the sentence as “Among all the SSL proteins, SSL10 is a well-studied member that has been found in most human and animal isolates of *S. aureus* and involved in several pathological process” (line 94-96).

8. *Line 124. Please justify selection of analysis of “SSL3, SSL7, SSL8, SSL10 and SSL11” but not other SSLs*

Response: We have added the reason for studying SSL3, 7, 8, 10, and 11 in the Results section: “To determine the cytotoxicity of SSLs on host cells, we cloned and expressed all of the 14 SSL members in *E. coli*. SSL3, SSL7, SSL8, SSL10, and SSL11 proteins were successfully purified with high quality for cytotoxicity assays. Human umbilical vein endothelial cells (HUVEC) were treated with these purified recombinant SSL proteins for different time periods. As determined by MTS assay (Fig. S1), SSL10, but not SSL3, SSL7, SSL8 or SSL11, significantly reduced the cell activity of HUVEC after 2-day treatment.” (line 113-119).

9. *Lines 143-145. Please perform statistical analysis of data points on Fig 1F and include results on the figure or include text that states complemented mutant was or was not significant different from the mutant strain.*

Response: Thanks for the suggestion. We have performed the statistical analysis as suggested, and found that compared to the *ssl10* knockout strain, *ssl10* complementation can significantly increase the LDH level. We have described this result as “*ssl10* knockout but not *ssl7* knockout significantly hampered the LDH release induced by *S.*

aureus supernatant, which can be rescued by *ssl10* complementation to a level similar to WT *S. aureus* (Fig. 1f).” in line 134-136, and we also labeled the *p* value in Fig. 1f.

10. Fig 4. Text in Fig 4F is too small. The Figure needs to be bigger.

Response: Thanks. We have revised Fig. 4f as suggested.

11. Line 235 “Furthermore, differ from other SSL proteins, several,” does not make sense. Please modify.

Response: Thanks. We changed the sentence to “Structural comparison of SSL10 with other SSL proteins (*i.e.*, SSL3, SSL4, SSL5, SSL7, SSL8, and SSL11) revealed that the SSL proteins share similar folds (Fig. S8b) but are quite different in electrostatic surface potential (Fig. 5b and Fig. S8c), which might be responsible for the diverse binding partners and functions of the SSL proteins.” (Line 224-228).

12. Line 237-238. “The distinct structural feature might endow SSL10 with specificity to interact with TNFR1.” This statement is not justified. Please justify or remove.

Response: Thanks for your suggestion. We have deleted the sentence.

13. Line 264-269. “The sequence identity between SSL10 and its homologs was relatively low, approximately 30-40% (Fig. S8). More significantly, the eight residues identified above in SSL10 differed from the corresponding residues in other SSL proteins on charge, while these corresponding residues were more similar among other

SSL proteins, indicating the specificity of SSL10 to interact with TNFR1ECD, which has more negatively charged surface.” These statements are not justified. It cannot be said that the difference in these 8 residues determines the TNFR1ECD specificity without mechanistic assessment when there is such extensive variation between the molecules. Likewise, there is no evidence provided that charge of molecules determines the TNFR1ECD specificity – this would require mutation of residues to opposite charges. Please revise, justify or remove.

Response: Thanks for your suggestion. We have deleted these statements.

14. Lines 279-282. “Therefore, the docking and mutagenesis analyses demonstrate that residues H64, K66, N85, S88, Q91, K206, K208, and Y209 are involved in SSL10 binding with TNFR1ECD and subsequent initiation of the necroptosis signal cascade.” This is not justified in the context of docking – please see above comments.

Response: Thanks for your suggestions. As described above in major comment #1 and #2, we have re-performed the docking analysis and analyzed the models according to your suggestions. The manuscript has been revised based on the new results (line 251-278). To avoid any overstatement of the proposed model, we used the word “the binding interface” instead of “key residues” when describing the interaction between SSL10 and TNFR1.

15. Line 307. Correct to HEK293.

Response: Thank you. We have corrected the typo.

REVIEWERS' COMMENTS:

Reviewer #3 (Remarks to the Author):

The authors have addressed all the major and minor points that I previously raised.